# Latent Principle Discovery for
# Language Model Self-Improvement

**Keshav Ramji**\*, **Tahira Naseem, Ramón Fernandez Astudillo**
IBM Research AI

## Abstract

When language model (LM) users aim to improve the quality of its generations, it is crucial to specify concrete behavioral attributes that the model should strive to reflect. However, curating such principles across many domains, even non-exhaustively, requires a labor-intensive annotation process. To automate this process, we propose eliciting these latent attributes that guide model reasoning toward human-preferred responses by explicitly modeling them in a self-correction setting. Our approach mines new principles from the LM itself and compresses the discovered elements to an interpretable set via clustering. Specifically, we employ a form of posterior-regularized Monte Carlo Expectation-Maximization to both identify a condensed set of the most effective latent principles and teach the LM to strategically invoke them in order to intrinsically refine its responses. We demonstrate that bootstrapping our algorithm over multiple iterations enables smaller language models (7-8B parameters) to self-improve, achieving +8-10% in AlpacaEval win-rate, an average of +0.3 on MT-Bench, and +19-23% in principle-following win-rate on IFEval. We also show that clustering the principles yields interpretable and diverse model-generated constitutions while retaining model performance. The gains that our method achieves highlight the potential of automated, principle-driven post-training recipes toward continual self-improvement.

## 1 Introduction

Modern language models (Grattafiori et al., 2024; OpenAI, 2024; DeepSeek-AI et al., 2025) have achieved striking fluency and coherence in open-ended generation, yet guiding them to satisfy multiple, possibly overlapping human-defined criteria remains a core challenge. Conventional approaches to align language models (LMs) rely on human annotations distinguishing between a chosen and rejected generation, even when their gap in quality may be nuanced and multi-faceted. Constitutional AI and other related paradigms (Bai et al., 2022b; Guan et al., 2025) consider a human-curated "constitution" of high-level attributes which models' responses should follow. While these frameworks enable models to be steered toward safer behavior, the static nature of their constitutions requires experts to anticipate nuances in advance and update rules manually as edge cases surface. As use cases proliferate, new failure modes arise – reliably synthesizing task-specific "amendments" and collecting annotations is a costly and time-consuming process – leading to brittleness and limited adaptability. We aim to automate the process of discovering these attributes for model improvement, obviating the need for human intervention or explicit domain adaptation.

Automatically discovering behaviors for self-improvement can be seen as a meta-level reasoning process. Recent efforts to induce reasoning capabilities in LMs have often focused on domains such as math and code where a gold reference answer exists and candidate answers are more easily verifiable (DeepSeek-AI, 2025). The availability of verifiable responses has also been capitalized for teaching self-correction (Kumar et al., 2025). However, in this work, we focus on open-ended text

---

\*Correspondence to `keshav.ramji@ibm.com`.

39th Conference on Neural Information Processing Systems (NeurIPS 2025).

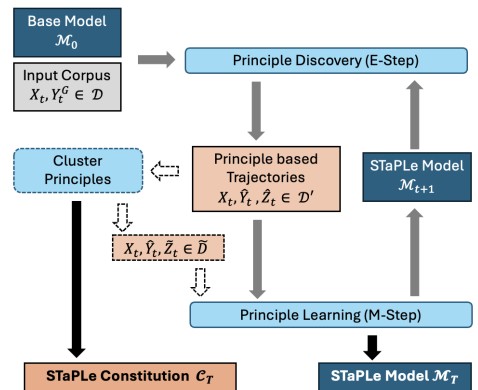
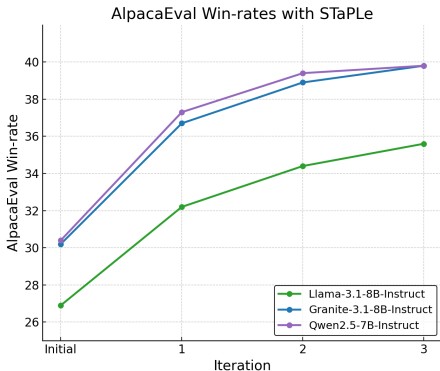

Figure 1: We introduce **Self-Taught Principle Learning** (**STaPLe**). (Left) Our Monte Carlo EM algorithm alternates between on-policy discovery and learning of latent principles guiding self-correction behavior. The principles may also be clustered to a compressed set, yielding human-interpretable constitutions $\mathcal{C}_t$ and models trained to follow them $\mathcal{M}_t$. (Right) The STaPLe algorithm induces self-improvement in AlpacaEval win-rate over three iterations for all three language models.

generation tasks that are challenging to verify; identifying situations for a human to intervene and induce a refined response can be especially tricky in such cases.

We introduce a novel approach to discover expressive principles, treating them as latent attributes in the self-correction setting to bridge an initial attempt and a target response. We find that the language model itself serves as an effective principle generator to improve its responses, contrasting prior works which rely on human-defined principles or strong model supervision. We design an Expectation-Maximization algorithm, **Self-Taught Principle Learning (STaPLe)**, which first leverages rejection sampling to identify principle candidates for self-correction and choose the induced refinement that is closest to the gold, then trains over these trajectories to learn this principle-guided refinement behavior. Iteratively applying this method results in a model trained on a dynamic constitution of elements produced from itself, implicitly learning the refinement goal to enable its self-correction abilities at inference-time. We also show that the discovered principles can be compressed to a smaller set for human readability by applying hierarchical clustering through posterior regularization, without compromising downstream performance.

We validate the efficacy of our method over several iterations on instruction-following benchmarks including MT-Bench (Zheng et al., 2023) and AlpacaEval (Li et al., 2023), and leverage Prometheus-v2.0 (Kim et al., 2024) to analyze win-rates with fine-grained, principle-following rubrics. Our results show that STaPLe outpaces baseline methods such as Self-Taught Reasoner (STaR; Zelikman et al. (2022)) (adapted for non-verifiable responses) and prompted refinement approaches like Self-Refine (Madaan et al., 2023). It continues to self-improve in performance over multiple iterations, before saturating. We also find that clustering largely matches or outperforms training on all principles.

Our key contributions can be summarized as follows:

- We propose a Monte Carlo Expectation-Maximization (EM) algorithm for iterative latent principle discovery and learning, to enable language model self-improvement.
- We find that on-policy generated principles are effective stimuli for self-correction in smaller LMs, and training to learn them improves the performance on MT-Bench, AlpacaEval-2.0.
- Clustering the set of discovered principles prior to training retains the performance gains of the full distribution while yielding an interpretable constitution.

## 2  Related Work

**Principle-Driven Language Modeling**    Early work in principle-driven alignment demonstrated that embedding high-level rule sets or "constitutions" into the training loop can steer model behavior without direct human labels for each generation. Constitutional AI (Bai et al., 2022b) introduced a

two-stage process in which a pre-trained model first generates critiques of its own outputs against a static constitution curated and synthesized a priori and learns from these critiques, then trains a preference model from on-policy data and performs RL. Dromedary (Sun et al., 2023) extended this idea by introducing Self-Align, an algorithm which generates prompts, applies a small set of human written principles with in-context learning, and then fine-tunes the model to learn the principle-guided responses; this was later extended by SALMON (Sun et al., 2024) to design an instructable, principle-following reward model (RM). ConstitutionalExperts (Petridis et al., 2024) and SPRI (Zhan et al., 2025) address the problem of mapping prompts to principles. Deliberative Alignment (Guan et al., 2025) introduces CoT reasoning over safety specifications, and trains models to learn these reasoning traces via SFT and online RL with a safety RM.

More recent efforts have sought to leverage models to draft and refine constitutions with limited human supervision. LEAP (Zhang et al., 2024) showed that models can propose new principles via self-critique given the gold response, synthesizing a list of task-specific principles that can be used for in-context learning. SAMI (Fränken et al., 2024) introduced a self-alignment mechanism where a strong model is used to generate a small set of principle candidates, and the target model is trained to maximize the mutual information between the constitution and the model's responses through an InfoNCE loss. IterAlign (Chen et al., 2024b) and ICAI (Findeis et al., 2025) also leverage strong frontier models such as GPT-4/GPT-4o (OpenAI, 2024) for constitution proposal. The former uses a red-teaming framework to identify model weaknesses, and uses the strong model to propose a principle towards helpfulness, harmlessness, and honesty; the latter considers principles as specifications over preference annotations, injecting them to reconstruct the annotation. Most recently, DeepSeek introduced a pointwise generative reward model (Liu et al., 2025) over self-generated principles, demonstrating that such an RM can be successfully used to improve inference-time scaling.

**Self-Correction and Self-Improvement**  The recent emergence of large reasoning models such as o3 and DeepSeek-R1 (OpenAI, 2025; DeepSeek-AI, 2025) has lead to a growing exploration into the ability of models to perform intrinsic refinement of their own outputs with internal feedback. However, much of the prior literature below either trains models over the improved responses alone or performs prompted self-refinement at inference-time, rather than *learning* this ability. STaR first leveraged model-generated critiques followed by revision, showing that alternating these two stages yields gains in instruction following (Zelikman et al., 2022). Self-Refine (Madaan et al., 2023) explored the setting of prompted multi-turn refinement: after producing an initial answer, the model is induced to critique itself, reflecting on the weaknesses of the current response on specific dimensions, proposing actionable changes, and reflecting them in the corrected response. ProMiSe (Ramji et al., 2024) extended this ability to smaller language models with weaker self-critique and refinement capabilities, showing that greedily refining responses relative to one attribute in a sequential manner improves performance, while demonstrating that training on synthetic dialogues modeling this self-refinement process improves dialogue question answering. APO (D'Oosterlinck et al., 2024) addressed the notion of minimally contrastive preference pairs, reinforcing the notion that revision along fewer attributes yields a better signal for preference optimization. Recently, reinforcement learning strategies have been explored to train models to directly perform self-correction. RISE (Qu et al., 2024) frames self-correction through a multi-turn Markov decision process (MDP), performs best-of-N sampling over sequential attempt rollouts, and uses offline RL to train the model to correct over these trajectories. SCoRe (Kumar et al., 2025) improves over this by a multi-turn RL formulation that boosts the quality of the initial attempt and leverages reward shaping to incentivize self-correction to improve the refined response. Beyond individual output corrections, recent work has shown that models can bootstrap their underlying capabilities in an iterative fashion over time. Several works suggest that sampling diverse responses or environmental interactions, filtering based on feedback or majority voting, and training on these on-policy generations can boost performance (Huang et al., 2023; Patel et al., 2024). SPIN (Chen et al., 2024c) introduced a self-play fine-tuning approach, wherein the model compares its generations against the ground-truth annotated responses in the SFT dataset yielding a preference pair, fine-tunes with a contrastive objective, and repeats this process iteratively. Huang et al. (2025) theoretically formalizes the self-improvement phenomenon through a "*sharpening*" process, wherein the model's policy moves towards maximizing its generations' self-reward.

**Latent Chain-of-Thought Learning**  Chain-of-thought (CoT) prompting (Wei et al., 2023; Kojima et al., 2022) elicits an explicit, verbalized step-by-step walkthrough of the reasoning trace guiding the model from the input to the final response. Simultaneously, STaR (Zelikman et al., 2022) leveraged a

gold response-conditioned rationalization of the CoT and fine-tuned the model to learn this reasoning behavior. The notion of rationalization as a latent variable modeling problem was previously explored in ELV (Zhou et al., 2020), under the framework of labeling examples with explanations through a variational EM algorithm. More recent approaches also treat chain-of-thought as a latent variable that may be trained over rather than purely to be induced at inference-time. TRICE (Phan et al., 2023) casts intermediate reasoning steps as latent codes, training the model to marginalize over them so that it internally develops coherent reasoning trajectories, through a Markov Chain Monte Carlo (MCMC) EM algorithm. LaTRO (Chen et al., 2024a) demonstrated that models can self-reward latent reasoning paths — generating candidate thought sequences, scoring them by task success, and reinforcing the most effective ones – through a variational framework. Concurrent work introduced BoLT (Ruan et al., 2025), showing that leveraging these implicit traces as supervision leads to gains in data efficiency and performance for continued pre-training on complex reasoning benchmarks by converting latent chain-of-thought into a self-supervised learning signal.

# 3 Self-Taught Principle Learning

We propose STaPLe, a self-improvement mechanism for (1) discovering human-interpretable principles by the model itself, aimed towards response revision, and (2) training language models to *intrinsically* invoke such principles and subsequently perform self-refinement of their generations (if needed) at inference time.

We view these principles as latent reasoning traces that bridge the gap between an initial model response and a reference target. In the vein of the Self-Taught Reasoner (STaR) (Zelikman et al., 2022), we leverage the gold response as a "hint" to propose principles and guide response refinement decisions. However, our formulation is generic and allows for the use of non-verifiable gold responses as hints. In particular, we use the proximity of the generated response to the reference response as a signal of correctness. Any similarity metric can be used to measure this proximity – the exact match metric, used for verifiable rewards, can be seen as one such instantiation.

Given a dataset $\mathcal{D} = \{(x_i, y_i^1, y_i^G)\}_{i=1}^n$, where $y_i^G$ is the gold response and $y_i^1$ is model's initial response for the $i^{th}$ sample, we aim to learn a latent response-improvement reasoning trace $z_i$ such that the probability of producing a response close to the gold reference is maximized. The latent reasoning trace, or *principle*, $z_i$ is also verbalized as natural language, i.e. discrete text tokens from vocabulary $\mathcal{V}^*$. We implement STaPLe to optimize the following marginal log-likelihood:

$$\mathcal{L}(\theta) = \log p(y^G \mid x, y^1) = \log \sum_{y^2 \in \mathcal{V}^*} \sum_{z \in \mathcal{V}^*} p(y^G \mid x, y^1, z, y^2) \cdot p(y^2, z \mid x, y^1; \theta) \qquad (1)$$

where $y^2$ is a model refinement of the initial response $y^1$ generated with aid of latent principle $z$. The distribution $p(y^G \mid x, y^1, z, y^2)$ acts as a prespecified parameter-free validator model indicating the probability of the current revision $y^2$ matching the gold response $y^G$. This can be implemented in many different ways, e.g. LLM-as-a-judge, but for the remainder of this work, it will be based on a string similarity function $f$, with the rule

$$p(y^G \mid x, y^1, z, y^2) \propto \begin{cases} f(y^2, y^G) - f(y^1, y^G), & \text{if } f(y^2, y^G) > f(y^1, y^G) \\ 0, & \text{otherwise.} \end{cases} \qquad (2)$$

We parametrize $p(y^2, z \mid x, y^1)$ by the language model itself, with parameters $\theta$. The gradient for the STaPLe objective is:

$$\begin{aligned}
\nabla_\theta \mathcal{L}(\theta) &= \nabla_\theta \log \sum_{y^2 \in \mathcal{V}^*} \sum_{z \in \mathcal{V}^*} p(y^G \mid x, y^1, z, y^2) \cdot p(y^2, z \mid x, y^1; \theta) \\
&= \sum_{y^2 \in \mathcal{V}^*} \sum_{z \in \mathcal{V}^*} \frac{p(y^G \mid x, y^1, z, y^2)}{p(y^G \mid x, y^1)} \nabla_\theta\, p(y^2, z \mid x, y^1; \theta) \\
&= \sum_{y^2 \in \mathcal{V}^*} \sum_{z \in \mathcal{V}^*} \frac{p(y^G \mid x, y^1, z, y^2) \cdot p(y^2, z \mid x, y^1; \theta)}{p(y^G \mid x, y^1)} \nabla_\theta \log p(y^2, z \mid x, y^1; \theta) \\
&= \mathbb{E}_{p(y^2, z \mid x, y^1, y^G)} \left\{ \nabla_\theta \log p(y^2, z \mid x, y^1; \theta) \right\} \qquad (3)
\end{aligned}$$

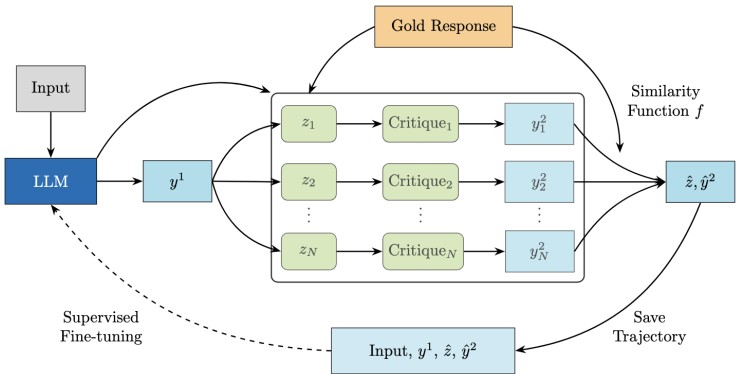

Figure 2: The figure above depicts the ***principle discovery*** (E-step) phase. We sample an initial response $y^1$ on-policy, then "hint" with the gold response to elicit candidate principles $z_{(1:N)}$. Then, we sample critiques on the initial response (only used in rejection sampling, and not included in the fine-tuning trajectories), which we use to obtain principle-guided refined responses $y^2_{(1:N)}$. The best refined response $\hat{y}^2$ is selected based on similarity to the gold response. We save the resulting trajectory, which is used for supervised fine-tuning in the ***principle learning*** (M-step) stage.

Optimizing this objective can be related to Expectation-Maximization (EM) learning and comprises the repeated application of two alternating stages: 1) discovery of the posterior distribution over principles and refinements (E-step), and 2) a principle learning stage (M-step). This is depicted in Figure 1. For details about the relationship to EM, see Appendix D.

In **principle discovery stage (E-step)**, we need to determine the posterior $p(y^2, z \mid x, y^1, y^G)$. However, obtaining the true posterior would require an intractable marginalization over $\mathcal{V}^*$. Instead, we use a Monte Carlo approximation of the expectation in the STaPLe objective for which we only need samples of this posterior. To sample from the posterior, one usual method is rejection sampling, which allows us to sample from any unnormalized score $g$

$$p(y^2, z \mid x, y^1, y^G) \propto g(y^2, z, x, y^1, y^G). \tag{4}$$

It rests to determine what is a good score $g$ to approximate the posterior samples. A straightforward option is using the numerator of the Bayes theorem. Here, we instead use a cycle consistency score

$$p(y^2, z \mid x, y^1, y^G; \theta) \propto p(y^G \mid x, y^1, z, y^2) \cdot \tilde{p}(y^2, z \mid x, y^1, y^G; \theta). \tag{5}$$

This allows us to use a proposal conditioned on the hint $y^G$, which can be expected to make the distribution of principles sharper and easier to learn, at the cost of bias. As with the Bayes theorem option, this uses the validator $p(y^G \mid x, y^1, z, y^2)$ to upweight samples that yield back the original gold answer $y^G$, given the refinement $y^2$ and the principle $z$. In this case, we still need to define a posterior proposal $\tilde{p}(y^2, z \mid x, y^1, y^G; \theta)$. One possibility is to just use the language model's policy $\pi_\theta$ at the current phase of training and prompt it to produce the principle and refinement. However, this bears the danger of collapsing into the trivial solution $y^2 = y^G$. To avoid this, we purposefully factor the principle-proposing distribution as

$$\tilde{p}(y^2, z \mid x, y^1, y^G; \theta) = p(y^2 \mid x, y^1, z; \theta) \cdot p(z \mid x, y^1, y^G; \theta) \tag{6}$$

so that sampling happens in two stages and the gold response is only seen by the principle generation stage [2]. Setting this distribution as the proposal, we have that a sample pair

$$z_n \sim p(z \mid x, y^1, y^G; \theta), \quad y^2_n \sim p(y^2 \mid x, y^1, z_n; \theta) \tag{7}$$

gets accepted with probability:

---

$$p_n = \frac{p(y^G \mid x, y^1, z_n, y_n^2)}{\max_{\substack{y \in \mathcal{V}^* z \in \mathcal{V}^*}} p(y^G \mid x, y^1, z, y^2)} \tag{8}$$

where we use a sample approximation for the maximum as in conventional rejection sampling strategies. We include a derivation of this rejection sampling rule in Appendix E.

In the practical implementation of the algorithm, there are two further aspects to consider. We include additional filtering where we compare the initial response $y^1$ to the gold reference $y^G$ using the similarity metric; if they are sufficiently close, we accept the response without further refinement and without sampling a $z$. Moreover, we only keep the sample with the highest $p(y^G \mid x, y^1, z_n, y_n^2)$ (i.e. Best-of-N) as opposed to keeping all accepted samples or using lower variance estimates such as Rao-Blackwellization. This can be seen as a form of hard-EM following the insight that Best-of-N can be seen as rejection sampling for a temperature $\to 0$. We provide further details in Appendix F.

The principle discovery stage yields a principle-augmented dataset $(x \cup y_1, z, y_2) \in \mathcal{D}'$. Note that if no refinements improve upon the initial generation relative to the gold response, we discard the sample; thus, the dataset $\mathcal{D}'$ only consists of those samples on which a principle improved the quality of the response towards the gold.

In the **principle learning stage (M-step)**, we use the data $\mathcal{D}'$ collected in the principle discovery stage for supervised fine-tuning of the language model. In particular, we train the model to maximize the log-likelihood of the refinement trajectories in $\mathcal{D}'$. The corresponding EM update can be written as:

$$\theta^{(t+1)} = \arg\max_{\theta} \mathbb{E}_{(x, y^1, z, \hat{y}^2) \in \mathcal{D}'} [\log p(y^2, z \mid x, y^1; \theta)] \tag{9}$$

Each optimization stage includes multiple gradient updates, with the data collected in the E-step being seen multiple times based on the number of epochs, after which a new batch of on-policy data is collected. The two stages are repeated, achieving incremental improvements till no further gains are seen with respect to the gold references or the maximum number of iterations is reached. Qualitatively, this should result in the fine-tuned LM being able to invoke principles conditioned on a prompt, and learning to produce high-quality responses conditioned on both the prompt and the invoked principle. We also draw a connection between STaPLe (this EM procedure) and variance-reduced self-play; this is discussed further in Appendix G.

### 3.1 Posterior Regularization via Clustering

To maximize the human interpretability of principles and their application relative to specific domains, it is beneficial to have a compressed set, or *constitution*, to distill to the model. However, the E-step described above, results in thousands of unique principles. We seek to project this set into a constrained subspace where the resulting principles serve as representatives for desirable attributes to be reflected. This can be achieved via posterior regularization (PR) in latent variable modeling. For a set of posteriors that satisfy the constraints $\mathcal{Q}$, the canonical posterior regularization framework solves the problem:

$$q^*(y^2, z \mid x, y^1, y^G) = \arg\min_{q \in \mathcal{Q}} \text{KL}(q(y^2, z \mid x, y^1, y^G) \,||\, p(y^2, z \mid x, y^1, y^G)) \tag{10}$$

which aims to project our unconstrained posterior into the space of posteriors that satisfy the constraints. From Ganchev et al. (2010), we obtain that the primal solution is given by:

$$\tilde{p}(y^2, z \mid x, y^1, y^G) \propto p(y^2, z \mid x, y^1, y^G) \cdot \exp\left(-\lambda g(z)\right) \tag{11}$$

where $g(z)$ are the constrained features, which only concerns the principles, and $\lambda$ is a Lagrange multiplier that must be set such that the expected value of the features under $\tilde{p}$ satisfies the constraints.

Consider the following cluster-based definition of the constraints: assume access to a clustering algorithm which yields a set of clusters $\{C_1, C_2, \dots C_K\}$. For each cluster, a representative element $\tilde{z}$ is chosen, forming the set $\tilde{Z} = \{\tilde{z}_1, \dots, \tilde{z}_K\}$. Now define $g(z) = \mathbf{1}(z \notin \tilde{Z})$ as a binary feature function. Thus, to ensure that the regularized posterior only places mass on the representative elements, we can enforce the constraint set $\mathcal{Q} = \{q : \mathbb{E}_q[g(z)] = 0\}$

For our case, the posterior regularized objective has a trivial solution of $\lambda \to \infty$ for elements not in the cluster representative set $\tilde{Z}$, which only leaves the problem of computing a new partition function.

Since we only need to sample from the posterior and we already do this through rejection sampling, retaining only the representative elements as a further stage of rejection sampling can be seen as a form of posterior regularization.

In particular, we consider hierarchical (agglomerative) clustering for several of its benefits: (1.) it requires no assumptions about number of clusters or cluster shape a priori, (2.) the algorithm is deterministic, ensuring that the same clusters would be obtained for a given configuration, and (3.) the algorithm is relatively fast, only taking a few seconds in practice over thousands of principles. To ensure that the clustering is performed in a semantically-aware manner, we first obtain a sentence embedding with an encoder-only model and perform clustering over these embeddings, consolidating principles that are lexically close.

Given the principle-augmented dataset $(x_i, y_i^1, z_i, y_i^2) \in \mathcal{D}'$ and a set $\tilde{Z}$ of cluster representative elements over $\mathcal{C} = \{C_i\}_{i=1}^k$, we aim to replace $\hat{z}_i$ with the element $\widetilde{z} \in \tilde{Z}$ that is closest in meaning to the original principle. Qualitatively, we want the set $\tilde{Z}$ to comprise the human-readable constitution, minimizing semantic overlap in its labels. We take the medoid as the cluster representative:

$$\widetilde{Z}_{medoid} = \{m_k : m_k = \arg \min_{m \in C_k} \sum_{j \in C_k} ||e_i - e_j||_2, k \in [1, K]\} \qquad \text{(Medoid Representatives)}$$

It suffices to retrieve the corresponding cluster $C_i$ for a sample $i$ and replace $z_i$ with $\tilde{z}_i \in \widetilde{Z}_{medoid}$. The resulting dataset from this augmentation, $(x_i, y_i^1, \tilde{z}_i, y_i^2) \in \widetilde{\mathcal{D}}$ is then used to train the model.

# 4 LMs Can Self-Improve with Iterative Principle Discovery

## 4.1 Experimental Setup

**Mixed-Domain Input Corpus.** We form a corpus of 100k samples for the principle discovery phase, consisting of four datasets: Anthropic HH-RLHF (Bai et al., 2022a), UltraFeedback (Cui et al., 2024), TL;DR (Stiennon et al., 2020), and HotpotQA (Yang et al., 2018), taken in equal proportion (i.e. 25k samples of each dataset, drawn randomly) and deduplicated by prompt. For all datasets, we use the existing, publicly-available human-annotated gold responses; for preference datasets, we take the chosen response to be the gold answer $y^G$. To run STaPLe, we use the first 50k samples for iteration 1, to heavily bootstrap off the first iteration, and then use 10k samples for each iteration thereafter, such that the input prompts are unseen for each iteration.

**Models and Hyperparameters.** We evaluate three performant small language models: Llama-3.1-8B-Instruct (Grattafiori et al., 2024; Meta, 2024), Granite-3.1-8B-Instruct (Granite Team, 2024; Granite Team and IBM, 2024), and Qwen2.5-7B-Instruct (Qwen, 2025). We also evaluate our method on Qwen3-32B to show that our approach scales to larger sizes (Appendix N). We use the all-MiniLM-L6-v2 model (Sentence Transformers, 2021) from SentenceTransformers as the embedding model to compute medoids in our clustering approach. We use the Rouge-L F1 score (Lin, 2004) to compare the similarity of candidate responses relative to the reference answer. We also include an ablation in Appendix P.2 using a prompted Phi-4 (Abdin et al., 2024) judge to score responses, leveraging additional compute to improve the quality of rejection sampling. We discuss all other major STaPLe algorithm and model training hyperparameters in Appendix C.

**Baselines.** We compared our method against several baselines in both the single-iteration and multi-iteration settings, in addition to the scores of each model's initial policy.

1. Prompted self-refinement to directly produce a self-critique and revision, akin to Self-Refine, without any principle or specific feedback criterion provided a priori.

2. Supervised fine-tuning on the gold responses of the first 50k samples in the mining corpus.

3. Following SCoRe, we adopt a STaR-like baseline for intrinsic self-correction; we apply the STaPLe algorithm and perform supervised fine-tuning on the best refined response (without principle-based refinement trajectory). This will henceforth be referred to as "STaR".

Table 1: Comparison of the STaPLe algorithm (unconstrained and constrained) against the baselines. The scores reported below are an average over 5 runs for all benchmarks. T1 and T2 refer to the first and second turns and WR is Win Rate.

| Model | MT-Bench (avg) | MT-Bench (T1) | MT-Bench (T2) | AlpacaEval | IFEval (WR) |
|---|---|---|---|---|---|
| **Llama-3.1-8B-Instruct** | | | | | |
| Initial Policy | 7.46 | 8.09 | 6.83 | 26.9 | – |
| Self-Refine | 7.40 | 8.05 | 6.75 | 26.1 | 51.2% |
| Gold-only SFT | 7.47 | 8.11 | 6.83 | 26.4 | 56.2% |
| STaR Iter 4 | 7.56 | 8.11 | 7.00 | 31.8 | 62.3% |
| STaPLe Iter 4 | **7.71** | **8.13** | **7.30** | 33.4 | 68.9% |
| Constrained STaPLe Iter 4 | 7.70 | **8.13** | 7.28 | **34.9** | **69.1%** |
| **Granite-3.1-8B-Instruct** | | | | | |
| Initial Policy | 7.83 | 8.59 | 7.08 | 30.2 | – |
| Self-Refine | 7.86 | 8.63 | 7.10 | 31.7 | 57.1% |
| Gold-only SFT | 7.86 | 8.68 | 7.05 | 30.1 | 55.8% |
| STaR Iter 4 | 7.96 | 8.68 | 7.25 | 35.6 | 62.1% |
| STaPLe Iter 4 | **8.04** | **8.69** | **7.41** | 38.4 | 67.6% |
| Constrained STaPLe Iter 4 | 8.03 | 8.65 | **7.41** | **38.8** | **68.4%** |
| **Qwen2.5-7B-Instruct** | | | | | |
| Initial Policy | 6.83 | 7.34 | 6.31 | 30.4 | – |
| Self-Refine | 6.91 | 7.41 | 6.40 | 30.7 | 58.4% |
| Gold-only SFT | 6.89 | 7.43 | 6.35 | 30.0 | 56.9% |
| STaR Iter 4 | 7.14 | 7.63 | 6.66 | 37.8 | 68.4% |
| STaPLe Iter 4 | **7.24** | **7.64** | **6.85** | **40.2** | **73.4%** |
| Constrained STaPLe Iter 4 | 7.22 | 7.60 | 6.84 | 39.9 | 72.1% |

We compare the STaPLe and STaR algorithms over four iterations – this is performed over the same number of samples per iteration, i.e. 50k samples in the first iteration and 10k samples for each subsequent one. Naturally, the other baselines are performed for a single iteration. We evaluate both the unconstrained (without clustering) and constrained (with clustering) versions of the STaPLe algorithm, and compare their performance.

**Evaluation.** We evaluate on the MT-Bench (Zheng et al., 2023) and AlpacaEval-2.0-LC (Li et al., 2023; Dubois et al., 2024) datasets, instruction-following evaluations designed to reflect alignment abilities of LLMs in chat settings; these are scored using the GPT-4o model (OpenAI, 2024). We also use the Prometheus-8x7B-v2.0 model (Kim et al., 2024) on the IFEval (Zhou et al., 2023) dataset, for fine-grained evaluation on principle-following rubrics, with additional experiments in Appendix I. At inference time, if a principle was invoked intrinsically given a prompt, the response is parsed to only score the refined generation, following the principle proposal – this is done similarly for the STaR baseline. Otherwise, we score the full generated response, and no special parsing is performed. For the IFEval results, the win-rate is with respect to the principle invoked – for example, if the principle is "Directness", the Prometheus judge assesses which response is more direct between the candidate generation and the generation from the initial policy. Given that the STaR baseline does not explicitly invoke a principle, we use the same principle invoked by the STaPLe model for that sample[3].

## 4.2 Results

**Latent Principle Learning Improves Response Quality.** The STaPLe algorithm outperforms the baselines on all benchmarks, across all models, as seen in Table 1. The MT-Bench average exceeds the best baseline by an average of +0.11 over the three models, with the Turn 2 increasing by an average of +0.22. The AlpacaEval win-rates improve over the initial policy by +5.3-7%, and improves over the STaR baseline by +1.6-2.8%. Furthermore, the IFEval win-rates on principle-following of the refined generations against the base policy using Prometheus improve by +5-6.6%. This suggests that training models to *explicitly invoke the principle* as an expressive form of a latent attribute is

---

[3]While the STaR-like baseline may not have seen this exact principle, it is valuable to analyze the effect of explicitly training on principle labels to see how well the final responses reflect those principles invoked.

Figure 3: Principle discovery rates of the STaPLe algorithm in the unconstrained (left) and constrained (right) settings. This represents the fraction of the trajectories saved from the principle discovery process (E-step) that contain a unique principle label that was unseen in previous iterations.

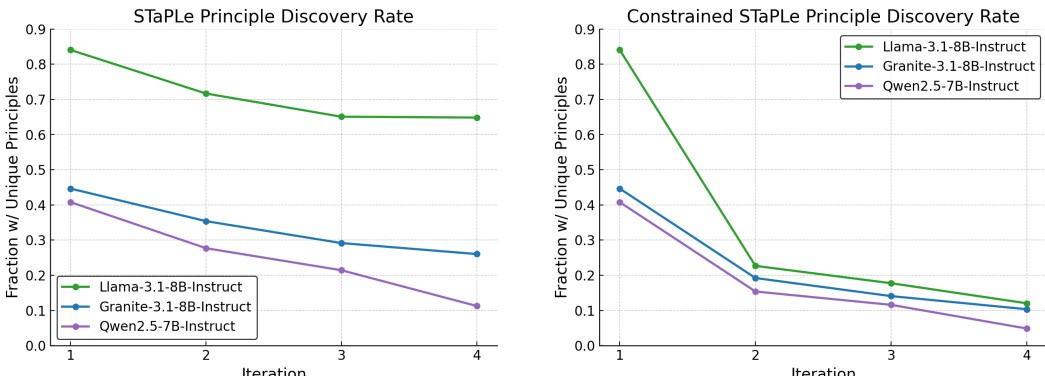

effective, as opposed to implicitly learning over this by simply training only on the refined responses (i.e. the STaR baseline). The Self-Refine baseline improves performance for the Granite and Qwen models, but not Llama-8B, suggesting that it is not as effective in zero-shot self-refinement without pre-identified principles. This corresponds with a higher IFEval win-rate for those models with strong self-refinement abilities.

**Iterative Principle Discovery Enables Self-Improvement.**  The results in Table 1 demonstrate the performance of our method in the fourth iteration of the Monte Carlo EM algorithm. Our algorithm outpaces the STaR baseline by a sizable margin throughout the execution of both algorithms; we include the full set of results in Appendix H. In iteration 3, the STaPLe scores outperform STaR and the initial policy on average across the three models by +0.16 and +0.29 on MT-Bench (avg.); +3.6% and +9.2% on AlpacaEval win-rate; and +7.9% and +21.0% on IFEval principle-following win-rate, respectively. We observe slight diminishing returns for Llama-8B and Granite-8B, as in iteration 4, the scores either remain at a similar level or drop slightly; however, Qwen-7B continues to improve on all three benchmarks. We further analyze principle-following quality in Appendix I and stepwise win-rates of iteration $t$ against iteration $t-1$ in Appendix J to reinforce the self-improvement induced by STaPLe. In Appendix K, we demonstrate that the model's intrinsic self-correction ability improves over the iterations. Moreover, we include an ablation in Appendix M showing that self-refinement improves when providing the constitution for the model to select principles from at inference-time.

**Clustering Balances Interpretability and Performance.**  In Table 1, we also include the performance of "constrained" STaPLe – the version of the algorithm with agglomerative clustering following the E-step during each iteration, and use the medoids of each cluster as a representative principle to yield dataset $\widetilde{\mathcal{D}}$. We find that this largely matches the performance of the "unconstrained" version, in fact outperforming it in AlpacaEval and IFEval win-rates for Llama-8B and Granite-8B. The full results can be found in Appendix P.1, where we ablate across different label replacement schemes (medoids, modes, and a perplexity-based method). For both versions, we observe a strong correlation in the trend (avg. $\rho \approx 0.95$-$0.96$) between the MT-Bench (avg.) and AlpacaEval results.

### 4.3  Analysis of Principle Discovery

It is also valuable to study the nature of the principle discovery process and the model-generated constitutions that we have aligned the language model toward. We include the full constitutions and perform more qualitative analysis on their distribution of elements when performing label replacement ("density" of the constitution) in Appendix L. In Figure 10, we show that the number of principles in the constitution under Constrained STaPLe decreases over the iterations, suggesting that the model converges to learning a relatively stable distribution of principles. In particular, the size of the constitution by iteration 4 is roughly 50% of the iteration 1 size, or even smaller.

This finding is reinforced by an analysis of the principle discovery rate – the fraction of refinement samples with new, unseen principles – in Figure 3. We show that this rate decreases over the iterations

under both the unconstrained and constrained versions of the STaPLe algorithm, suggesting that all models learn to re-use principles accordingly. The observation that constrained STaPLe helps to accelerate this convergence to a condensed set of principles reinforces the motivation behind the introduction of clustering as a posterior regularization mechanism. This also highlights one of the advantages of using the LM to approximate the posterior distribution, as the changing nature of the learned posterior can be observed over the iterations and elicited via on-policy sampling.

## 5  Discussion and Limitations

The STaPLe algorithm guides a largely autonomous self-improvement process, with the exception of a few hyperparameters that are to be set, discussed in Appendix C. As a result, the algorithm does not require human supervision beyond the labels in the curated (publicly-available) mining corpus. This work focuses primarily on smaller, non-reasoning language models, which struggle with self-refinement. Nonetheless, we find that STaPLe is also effective for the Qwen3-32B reasoning model to self-improve (see Appendix N), an encouraging sign for further scaling its applications.

While the distribution of principles is mined relative to the mining corpus' task distribution, the STaPLe algorithm itself is task-agnostic, and can be used for any distribution of datasets where a reliable gold reference exists, or with paired preference data. However, designing a task-aware version of the STaPLe algorithm may reveal further insights into the model's task-dependent self-correction mechanisms while inducing a curriculum. In this work, we focus on the two-turn self-correction setting – at the same time, interesting insights could be extracted regarding the compositional nature of principles when extending to further refinement attempts, which also yields more diverse trajectories (combinatorially many possible), even over a condensed set of principles.

A core aim of alignment research is to balance human-readability with machine-readability. The STaPLe algorithm succeeds in achieving this by discovering principles that are useful to the model for self-correction, while compressing them to a smaller set via clustering for a human reader to analyze. We believe that this work and the notion of LM self-improvement keeps with the theme of the Bitter Lesson (Sutton, 2019), when facilitated in a relatively autonomous fashion. Specifically, we aim to limit the influence of human-driven priors or constraints on the algorithm; this is reflected further by our clustering technique, and our ablation in Appendix P.3 to fully automate this as well. At the same time, we acknowledge the value of human oversight on the alignment process; as such, we believe that human-in-the-loop analysis of the principles as a post-processing mechanism following the E-step of each iteration would be valuable to further mitigate misalignment risks or potentially harmful principles (see Appendix R).

Our work introduces principles as an expressive form of a latent chain-of-thought, relying on the power and scale of pre-training to bootstrap our mechanism in post-training. One can view our constitutions as general-purpose abstractions that inform generation quality – such a notion can be easily extended to new domains, such as difficult reasoning, or for multi-aspect verification. It provides a concrete verbalization of the dimensions along which Pareto improvement is desirable to maximize instruction-following capabilities. Moreover, they can serve to guide further training, including online reinforcement learning with STaPLe as a warm-up, and integration with memory frameworks for continually discovering and revising target attributes. The evidence that the discovered constitutions are useful both in iteratively training LMs for self-correction as well as through their inference-time application (see Appendix M) presents a promising outlook for this research direction.

## 6  Conclusion

We introduced a new language model self-improvement method which uses model-generated latent principles to learn intrinsic self-correction. These serve a purpose akin to a reasoning chain-of-thought, boosting the quality of LM generations on alignment-focused benchmarks. Furthermore, our posterior-regularized Monte Carlo EM algorithm shows that the model can continue to improve over multiple iterations, while simultaneously compressing the principles to a human-interpretable constitution. We also show that our clustering approach balances performance with the diversity of the generated constitution, thus adding valuable utility to the STaPLe algorithm. The efficacy of STaPLe highlights the potential for constitutional alignment with self-generated principles to improve model responses in an interpretable manner with minimal human supervision.

**Acknowledgments.** The authors would like to thank Sara Rosenthal, Yikang Shen, Radu Florian, and Salim Roukos for valuable input and feedback during this work. KR would also like to thank Brian Williams and Arduin Findeis for helpful discussions related to this work and the impacts of principle discovery, and Sriram Tolety for feedback on a draft of this paper.

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

# Appendix

# Table of Contents

# A  Formal Description of STaPLe Algorithm

We provide a full, formal description of the STaPLe algorithm below. We use $y^1$ and $y^2$ notationally to avoid confusion with the sample indices. We use general variables for components which may be ablated on: the similarity function $f$, clustering algorithm $\mathcal{C}$ and label replacement scheme $\mathcal{R}$. We leave the $M$-step in terms of the dataset $D'$ for generality, although if clustering were to be performed, one would use $\widetilde{\mathcal{D}}$ instead.

---

**Algorithm 1** Self-Taught Principle Learning (STaPLe)

---

**Require:** Dataset $\mathcal{D} = \{(x_i, y_i^G)\}_{i=1}^n$, pretrained LM parameters $\theta^{(0)}$, number of EM iterations $T$, number of principle samples $N$, similarity threshold $\tau$, similarity function $f(\cdot, \cdot)$; (optional) embedding model EMB, clustering algorithm $\mathcal{C}$, and cluster representative scheme $\mathcal{R}$

1: **for** $t = 0, \ldots, T - 1$ **do**
2:     **E-step:** initialize $\mathcal{D}' \leftarrow \varnothing$.
3:     **for** each $(x_i, y_i^G) \in \mathcal{D}$ **do**
4:         Sample initial response $y_i^1 \sim \pi_{\theta^{(t)}}(\cdot \mid x_i)$.
5:         **if** $f(y_i^1, y_i^G) < \tau$ **then**                            ▷ needs refinement
6:             Draw principles $\{z_i^{(j)}\}_{j=1}^N \sim p_{\theta^{(t)}}(z \mid x_i, y_i^1, y_i^G)$.
7:             **for** $j = 1, \ldots, N$ **do**
8:                 Generate critique $c_i^{(j)} \leftarrow \text{Critique}(y_i^1, z_i^{(j)})$.
9:                 Sample refinement $y_i^{2,(j)} \sim \pi_{\theta^{(t)}}(\cdot \mid x_i, y_i^1, z_i^{(j)}, c_i^{(j)})$.
10:             **end for**
11:             $j^* \leftarrow \text{argmax}_j \; f(y_i^{2,(j)}, y_i^G)$
12:             $(z_i, y_i^2) \leftarrow (z_i^{(j^*)}, y_i^{2,(j^*)})$
13:             **if** $f(y_i^2, y_i^G) > f(y_i^1, y_i^G)$ **then**
14:                 Add trajectory $(x_i, y_i^1, z_i, y_i^2)$ to $\mathcal{D}'$.
15:             **end if**
16:         **end if**
17:     **end for**
18:     **(Optional):** Cluster the principles to a smaller set in augmented dataset $\widetilde{D}$
        Clusters $C \leftarrow \mathcal{C}(\text{EMB}(\{z_i\}))$
        Assign cluster representatives (e.g. Medoid) $\widetilde{Z} \leftarrow \mathcal{R}(C)$
        Augment dataset $\widetilde{\mathcal{D}} \leftarrow \text{Rep}(\mathcal{D}', \widetilde{Z})$
19:     **M-step:**
$$\theta^{(t+1)} \leftarrow \arg\max_\theta \sum_{(x, y^1, z, y^2) \in \mathcal{D}'} \log p_\theta(y^2, z \mid x, y^1).$$

20: **end for**
**Ensure:** Final LM parameters $\theta^{(T)}$

---

# B  Reproducibility Statement

In addition to the algorithm description above (Algorithm 1) and experimental details in Section 4.1, we include the hyperparameters used and model training details in Appendix C and the prompts used in the STaPLe algorithm in Appendix O. We make all evaluation results available in tabular format throughout the main paper and the appendices for comparability. We also will publicly release the code for the STaPLe algorithm, to further facilitate reproducibility of our self-improvement method.

# C  STaPLe Hyperparameters and Training Details

**STaPLe Algorithm Hyperparameters.** We use a Rouge-L F1 threshold of 0.4 for the similarity threshold ($f(y, y^G)$ – if the initial response exceeds this threshold, we do not pursue refinement). For the ablation using a Phi-4 judge in Appendix P.2, the threshold was set to be 9 (on a scale of 1-10). The other major hyperparameters involved in the execution of the STaPLe algorithm are $N$, the number

of principles to sample, and the distance threshold for the clustering algorithm. STaPLe requires an inference time budget of $3N + 1$ for the Rouge-L version, and $4N + 2$ for the LLM-as-a-judge version – we set $N = 16$ to balance runtime per iteration of the algorithm with sufficient exploration of diverse principles. During principle discovery, we sample principles, critiques, and responses at a temperature of $0.7$; the maximum number of tokens for principle proposal and critique is set at $500$, and is set at $1024$ for the refined response. We use $4 \times$H100 Nvidia GPUs for the principle discovery phase, with a separate vLLM (Kwon et al., 2023) instance per GPU.

We set a distance threshold $\delta$ to avoid setting a specific target number of clusters when performing agglomerative clustering. The current results involve manually setting a distance threshold, where the authors analyzed the resulting set of clusters and for each of the first three iterations, ensured that there are at least 30 clusters. Fortunately, given the speed of agglomerative clustering, this is fairly easy to do. For the first iteration, the Euclidean distance thresholds were set at 8 (Llama and Qwen) and 6 (Granite); for iterations 2-4, the thresholds were decreased to 7 and 5, respectively. Alternatively, one could automate this process by designing an objective over the diversity (semantic or surface-level) of the cluster medoid labels and performing a hyperparameter search. This is explored further in Appendix P.3 using Bayesian hyperparameter optimization tools to identify an appropriate, model-specific distance threshold. The threshold $\tau_{PPL}$ for the perplexity difference label-replacement scheme, described in Appendix P.1, was set at $0.2$.

**Model Training.** We perform full supervised fine-tuning for 3 epochs at a learning rate of $1 \times 10^{-6}$ with the AdamW optimizer (Loshchilov and Hutter, 2019), with a sequence length of 4096. All experiments were performed on $8 \times$H100 Nvidia GPUs.

# D   Derivation of the Monte Carlo EM Gradient

Recall that the conditional log-likelihood is defined as:

$$\mathcal{L}(\theta) = \log p(y^G \mid x, y^1) = \log \sum_{y^2 \in \mathcal{V}^*} \sum_{z \in \mathcal{V}^*} p(y^G \mid x, y^1, z, y^2) \cdot p(y^2, z \mid x, y^1) \qquad (12)$$

The usual EM formulation is given from the Variational Lower Bound (ELBO):

$$
\begin{aligned}
\mathcal{L}(\theta) &= \log p(y^G \mid x, y^1) \\
&= \log \sum_{y^2 \in \mathcal{V}^*} \sum_{z \in \mathcal{V}^*} p(y^G \mid x, y^1, z, y^2) \cdot p(y^2, z \mid x, y^1) \\
&= \log \sum_{y^2 \in \mathcal{V}^*} \sum_{z \in \mathcal{V}^*} q(z, y^2 \mid x, y^1) \cdot \frac{p(y^G, y^2, z \mid x, y^1)}{q(z, y^2 \mid x, y^1)} \\
&\geq \sum_{y^2 \in \mathcal{V}^*} \sum_{z \in \mathcal{V}^*} q(z, y^2 \mid x, y^1) \cdot \log \left( \frac{p(y^G, y^2, z \mid x, y^1)}{q(z, y^2 \mid x, y^1)} \right) \\
&= \log p(y^G \mid x, y^1) - \mathrm{KL}(q \,||\, p(y^2, z \mid x, y^1, y^G)) \\
&= \mathbb{E}_{q(z, y^2 \mid x, y^1)} \left\{ \log p(y^G \mid x, y^1, z, y^2) + \log p(y^2, z \mid x, y^1) \right\} + \mathrm{H}(q). \qquad (13)
\end{aligned}
$$

From the fourth line it is clear that the bound is tight only if the KL is zero i.e. the variational distribution is equal to the posterior over the latent

$$q(z, y^2 \mid x, y^1) = p(y^2, z \mid x, y^1, y^G). \qquad (14)$$

Once this is determined (E-step) it rests to optimize the last line (M-step). For a single gradient update, this yields the STaPLe update rule

$$\nabla_\theta \mathcal{L}(\theta) = \mathbb{E}_{p(y^2, z \mid x, y^1, y^G)} \left\{ \nabla_\theta \log p(y^2, z \mid x, y^1; \theta) \right\}. \qquad (15)$$

Since we do not derive a closed form solution for $p(y^2, z \mid x, y^1, y^G)$ but rather we only obtain samples from it, this algorithm is better understood as related to Monte Carlo EM (Wei and Tanner, 1990).

## E  Derivation of Rejection Sampling Rule

We here refresh rejection sampling and show the corresponding STaPLe formulas. Rejection sampling or accept/reject sampling (von Neumann, 1951) allows to sample from distributions with an intractable partition function

$$p(y^2, z \mid x, y^1, y^G) = \frac{g(y^2, z, x, y^1, y^G)}{Z} \tag{16}$$

by sampling from a proposal distribution $q(y^2, z)$ and accepting the sample with probability

$$p_n = \frac{1}{M} \cdot \frac{p(y^2, z \mid x, y^1, y^G)}{q(y^2, z)} \tag{17}$$

where

$$M = \max_{y^2 \in \mathcal{V}^* z \in \mathcal{V}^*} \frac{p(y^2, z \mid x, y^1, y^G)}{q(y^2, z)} \tag{18}$$

scales the ratio to ensure that it is a valid probability. Note $Z$ cancels out when replacing the last equation into the previous, making the probability not dependent from the partition function.

For STaPLe we have defined

$$g(y^2, z, x, y^1, y^G) = p(y^G \mid x, y^1, z, y^2) \cdot \tilde{p}(y^2, z \mid x, y^1, y^G) \tag{19}$$

and

$$q(y^2, z) = \tilde{p}(y^2, z \mid x, y^1, y^G) \tag{20}$$

leading to

$$
\begin{aligned}
p_n &= \frac{1}{\max\limits_{y \in \mathcal{V}^* z \in \mathcal{V}^*} \frac{p(y^G \mid x, y^1, z, y^2) \cdot \tilde{p}(y^2, z \mid x, y^1, y^G)}{Z \cdot \tilde{p}(y^2, z \mid x, y^1, y^G)}} \\
&\quad \cdot \frac{p(y^G \mid x, y^1, z, y^2) \cdot \tilde{p}(y^2, z \mid x, y^1, y^G)}{Z \cdot \tilde{p}(y^2, z \mid x, y^1, y^G)} \\
&= \frac{p(y^G \mid x, y^1, z_n, y_n^2)}{\max\limits_{y \in \mathcal{V}^* z \in \mathcal{V}^*} p(y^G \mid x, y^1, z, y^2)}
\end{aligned}
\tag{21}
$$

## F  Hard EM and Best-of-N in Monte Carlo EM

Hard Expectation Maximization (Dempster et al., 2018) refers to EM algorithms where the posterior over the latents is replaced by the greedy most probable output of the posterior e.g. obtained via Viterbi decoding for classical Hidden Markov Model latent variable models. In other words, we collapse the posterior over its most likely output. We can use the result in (Liu et al., 2024, S 3.2) to prove that this coincides with Best-of-N choice according to the score $p(y^G \mid x, y^1, z_n, y_n^2)$ under conditions that our algorithm meets.

For this, we consider our score-based distributions with an additional temperature term $\tau$ in our score

$$p(y^2, z \mid x, y^1, y^G) = \frac{g(y^2, z, x, y^1, y^G)^{\frac{1}{\tau}}}{Z} \tag{22}$$

It is self-evident that for $\tau \to 0$ this distribution collapses over its most likely value. For the values set in our algorithm (see Appendix E), and also using this temperature in the proposal, we get

$$p_n = \exp\left(\log p(y^G \mid x, y^1, z_n, y_n^2)^{\frac{1}{\tau}} - \max_{y \in \mathcal{V}^* z \in \mathcal{V}^*} \log p(y^G \mid x, y^1, z, y^2)^{\frac{1}{\tau}}\right). \qquad (23)$$

As long as we approximate max using our set of samples, as is customary in rejection sampling, there will always be one output with probability 1. This is true even if $\tau \to 0$, which implies that, under these conditions, Monte Carlo Hard EM can be implemented by simply picking the $\arg\max$ of the score.

## G   Self-Play Equivalence

The STaPLe Monte Carlo EM approach can equivalently be described through the lens of *self-play*, somewhat akin to SPIN (Chen et al., 2024c). That is, we can formulate a two-player game wherein the adversary produces a response, and the agent's role is to 1. produce a revised response to the prompt that improves over the adversary's generation relative to the gold, and 2. specify the dimension or aspect on which it improved over the adversary. In the first iteration, we take the same LM to play the both roles. In subsequent iterations, given the policy $\pi_\theta$ has now learned self-correction behavior, we take the initial response (opponent's generation) as the starting point, which we posit to be similar to generations sampled from the base policy $\pi_0$ – that is, $y^a \sim \pi_0(\cdot \mid x) \approx y^b \in (y^b, z, y^c) \sim \pi_\theta(\cdot \mid x)$. At the same time, the agent's policy updates to $\pi_\theta$, which learns principle-conditioned self-refinement, thus improving the agent's ability to perform its primary objectives.

Formally, we can define the self-play advantage of the refinement over the adversary's generation as

$$A(y^2, y^1; x, y^G) = f(y^2, y^G) - f(y^1, y^G)$$

Recall that in the STaPLe algorithm, if the agent "loses" – that is, it fails to produce a refinement that improves over the initial response – the sample is discarded. The nature of the advantage depends on the instantiation of the similarity function $f$; for instance, under exact match, this collapses to a binary indicator. The objective in the self-correction setting is to maximize the expected advantage under $\pi_\theta$:

$$J(\theta) = \mathbb{E}_{y^1, z, y^2 \sim \pi_\theta} [A(y^2, y^1; x, y^G)]$$

The score-function gradient is thus:

$$\nabla_\theta J(\theta) = \mathbb{E}_{y^1, z, y^2 \sim \pi_\theta} [A(y^2, y^1; x, y^G) \nabla_\theta \log \pi_\theta(y^1, z, y^2 \mid x)] \qquad (24)$$

Given that $y^1$ is treated as being sampled from a fixed policy, we can write this gradient in terms of $z$ and $y_2$:

$$\nabla_\theta J(\theta) = \mathbb{E}_{(z, y^2) \sim \pi_\theta} [A(y^2, y^1; x, y^G)[\nabla_\theta \log \pi_\theta(z \mid x, y^1) + \nabla_\theta \log \pi_\theta(y^2 \mid x, y^1, z)]] \qquad (25)$$

**Theorem 1** (Equivalence of EM and Self-Play Gradients). *Assume the setting of an input $x$, an initial model response $y^1 \sim \pi_\theta(\cdot \mid x)$, a latent principle $z \sim \pi_\theta(\cdot \mid x, y^1, y^G)$, and a refinement $y^2 \sim \pi_\theta(\cdot \mid x, y^1, z)$. Then, the EM gradient for the STaPLe algorithm is equivalent to the REINFORCE score-function gradient under variance-reduced self-play, given by Equation 25, under the the self-play advantage and the validator assignment*

$$v(y^2 \mid x, y^1) = \mathbf{1}(f(y^2, y^G) > f(y^1, y^G))$$

*Proof.* Define the E-step posterior over $(y^2, z)$, with parameters $\bar{\theta}$ (held constant in the M-step) as

$$p_{\bar{\theta}}(y^2, z \mid x, y^1, y^G) \propto p(y^G \mid x, y^1, y^2) \cdot \pi_{\bar{\theta}}(z \mid x, y^1, y^G) \cdot \pi_{\bar{\theta}}(y^2 \mid x, y^1, z) \qquad (26)$$

Consider the unnormalized validator $v(y^2 \mid x, y^1) \propto p(y^G \mid x, y^1, y^2)$. Then, by Bayes' theorem:

$$p_{\bar{\theta}}(y^2, z \mid x, y^1, y^G) = \frac{v(y^2 \mid x, y^1) \cdot \pi_{\bar{\theta}}(z \mid x, y^1, y^G) \cdot \pi_{\bar{\theta}}(y^2 \mid x, y^1, z)}{Z_{\bar{\theta}}(x, y^1, y^G)} \qquad (27)$$

where $Z_{\bar{\theta}}(x, y^1, y^G) = \sum\limits_{z, y^2} v(y^2 \mid x, y^1) \cdot \pi_{\bar{\theta}}(z \mid x, y^1, y^G) \cdot \pi_{\bar{\theta}}(y^2 \mid x, y^1, z)$ only depends on $\bar{\theta}$, not on $\theta$.

The M-step maximizes the expected log-likelihood under the fixed E-step posterior, that is:

$$\mathcal{L}(\theta) = \mathbb{E}_{(y^2, z) \sim p_{\bar{\theta}}(\cdot \mid x, y^1, y^G)}[\log \pi_\theta(z \mid x, y^1, y^G) + \log \pi_\theta(y^2 \mid x, y^1, z)] \tag{28}$$

Therefore, the M-step gradient is:

$$\nabla_\theta \mathcal{L}(\theta) = \mathbb{E}_{(y^2, z) \sim p_{\bar{\theta}}}[\nabla_\theta \log \pi_\theta(z \mid x, y^1, y^G) + \nabla_\theta \log \pi_\theta(y^2 \mid x, y^1, z)] \tag{29}$$

Next, we consider the assignment of the EM validator to be in terms of the comparison between initial response $y^1$ and refined response $y^2$ with respect to $y^G$ over the similarity function $f$. That is, take $v(y^2 \mid x, y^1) = \mathbf{1}(f(y^2, y^G) > f(y^1, y^G))$; such that we only accept refinements that beat $y^1$ under $f(\cdot, y^G)$. This reflects the STaPLe algorithm's accept/reject criterion. Substituting for $p_{\bar{\theta}}$ yields:

$$\nabla_\theta \mathcal{L}(\theta) \propto \mathbb{E}_{y^2, z \sim \pi_{\bar{\theta}}} [\mathbf{1}(f(y^2, y^G) > f(y^1, y^G))[\nabla_\theta \log \pi_\theta(z \mid x, y^1, y^G) + \nabla_\theta \log \pi_\theta(y^2 \mid x, y^1, z)]] \tag{30}$$

with the normalization constant $\frac{1}{Z_{\bar{\theta}}(x, y^1)}$ being constant with respect to $\theta$ in the M-step.

This is consistent with the score-function gradient defined in Equation 25, under the indicator definition. Note that under practical variance reduction, all credit is attributed to the final refinement, with none attributed to the principle. As such, we can rewrite this gradient as:

$$\nabla_\theta \mathcal{L}(\theta) \propto \mathbb{E}_{y^2, z \sim \pi_{\bar{\theta}}} [\mathbf{1}(f(y^2, y^G) > f(y^1, y^G))[\nabla_\theta \log \pi_\theta(y^2 \mid x, y^1, z)]] \tag{31}$$

As a corollary, to generalize to real-valued rewards such as Rouge-L, reward models, or LLM-as-a-judge scores, we instead replace this hard indicator with an advantage function $A(y^2, y^1; x, y^G) = f(y^2, y^G) - f(y^1, y^G)$. Concretely, this implies taking the validator $v(y^2 \mid x, y^1) \propto e^{A(y^2, y^1; x, y^G)}$.

In the canonical REINFORCE self-play setting (Dayan, 1990; Sutton, 1984), the reward $R(\tau)$ over the trajectory $\tau$ is often replaced by an advantage to reduce the variance of the Monte Carlo estimate, introducing $A(\tau) = R(\tau) - b$ for a comparison $b$. This yields a gradient $\nabla_\theta J(\theta) = \mathbb{E}[A(\tau) \nabla_\theta \log[\pi_\theta(\tau)]]$ in practice. In our setting, we are simply taking the score of the initial response $f(y^1, y^G)$ to be the comparison.

Performing this substitution in the current form of the EM gradient yields:

$$\nabla_\theta \mathcal{L}(\theta) \propto \mathbb{E}_{y^2, z \sim \pi_\theta} [A(y^2, y^1; x, y^G) \nabla_\theta \log \pi_\theta(y^2 \mid x, y^1, z)]] \tag{32}$$

This recovers Equation 25, the self-play REINFORCE gradient, concluding the proof.

$\square$

# H    Complete Table: Self-Improvement over Multiple Iterations

In this section, we include the complete tables over four iterations of the STaPLe algorithm, to demonstrate the model's progression of self-improvement. As shown in Table 2, STaPLe outpaces the STaR baseline by a substantial margin throughout the execution of both algorithms, even in spite of the improvements of Llama-8B and Granite-8B saturating by the end of iteration 3. While both algorithms have fairly similar MT-Bench Turn-1 scores by iteration 4, the Turn-2 score is substantially higher (average of +0.22) for STaPLe. We observe similar general trends for the models in AlpacaEval win-rate and Prometheus-based IFEval principle-following win-rate, as well.

Table 2: Self-improvement over four iterations of the STaPLe algorithm, compared against the STaR baseline (SFT without the principle operating as a latent CoT between the initial and refined attempts). Note that the SFT sample counts for iterations 2-4 differ as the principles are discovered by different models – the STaR Iter 1 and STaPLe Iter 1 models, respectively. Numbers in parentheses denote training set size, based on the number of samples which successfully refined.

| Model | MT-Bench (avg) | MT-Bench (T1) | MT-Bench (T2) | AlpacaEval | IFEval WR |
|---|---|---|---|---|---|
| **Llama-3.1-8B-Instruct** | | | | | |
| Initial Policy | 7.46 | 8.09 | 6.83 | 26.9 | – |
| STaR Iter 1 (28.2k) | 7.43 | 8.04 | 6.81 | 29.1 | 55.5% |
| STaPLe Iter 1 (28.2k) | 7.66 | 8.15 | 7.16 | 32.2 | 65.6% |
| STaR Iter 2 (6.0k) | 7.47 | 8.08 | 6.86 | 30.6 | 57.7% |
| STaPLe Iter 2 (6.1k) | 7.74 | 8.19 | 7.29 | 34.4 | 66.2% |
| STaR Iter 3 (6.1k) | 7.51 | 8.10 | 6.91 | 31.5 | 61.0% |
| STaPLe Iter 3 (6.3k) | **7.74** | **8.16** | **7.31** | **35.6** | 68.8% |
| STaR Iter 4 (6.3k) | 7.56 | 8.11 | 7.00 | 31.8 | 62.3% |
| STaPLe Iter 4 (6.6k) | 7.71 | 8.13 | 7.30 | 33.4 | **68.9%** |
| **Granite-3.1-8B-Instruct** | | | | | |
| Initial Policy | 7.83 | 8.59 | 7.08 | 30.2 | – |
| STaR Iter 1 (24.1k) | 7.83 | 8.61 | 7.05 | 33.0 | 57.3% |
| STaPLe Iter 1 (24.1k) | 7.99 | 8.69 | 7.29 | 36.7 | 65.1% |
| STaR Iter 2 (5.4k) | 7.86 | 8.63 | 7.10 | 34.7 | 59.5% |
| STaPLe Iter 2 (5.2k) | 8.04 | 8.74 | 7.34 | 38.9 | 65.2% |
| STaR Iter 3 (5.9k) | 7.92 | 8.66 | 7.18 | 35.4 | 61.9% |
| STaPLe Iter 3 (5.9k) | **8.06** | **8.75** | 7.38 | **39.8** | **71.6%** |
| STaR Iter 4 (6.2k) | 7.96 | 8.68 | 7.25 | 35.6 | 62.1% |
| STaPLe Iter 4 (6.3k) | 8.04 | 8.69 | **7.41** | 38.4 | 67.6% |
| **Qwen2.5-7B-Instruct** | | | | | |
| Initial Policy | 6.83 | 7.34 | 6.31 | 30.4 | – |
| STaR Iter 1 (30.9k) | 6.85 | 7.39 | 6.31 | 34.5 | 61.0% |
| STaPLe Iter 1 (30.9k) | 7.03 | 7.48 | 6.59 | 37.3 | 68.2% |
| STaR Iter 2 (6.5k) | 6.98 | 7.45 | 6.51 | 36.9 | 63.0% |
| STaPLe Iter 2 (6.5k) | 7.14 | 7.55 | 6.73 | 39.4 | 66.2% |
| STaR Iter 3 (7.1k) | 7.08 | 7.58 | 6.59 | 37.6 | 66.4% |
| STaPLe Iter 3 (7.0k) | 7.20 | 7.63 | 6.78 | 39.8 | 72.5% |
| STaR Iter 4 (7.1k) | 7.14 | 7.63 | 6.66 | 37.8 | 68.4% |
| STaPLe Iter 4 (7.1k) | **7.24** | **7.64** | **6.85** | **40.2** | **73.4%** |

# I Prometheus Win-rates on MT-Bench and AlpacaEval

Given that in Section 4.2, we have sampled responses with intrinsic principle-conditioned self-correction behavior from the language model for MT-Bench and AlpacaEval, we can further study the quality of the Prometheus-8x7B-v2.0 model in producing judgements over a fine-grained rubric. We specifically would like to understand whether the model's responses – which, as per Tables 1 and 2, achieve improvements in score – actually reflect the principles they invoke.

This method corresponds to the IFEval principle-following win-rates reported in Section 4.2. As such, note that the AlpacaEval win-rate in this section differs from the standard AlpacaEval scoring – this is the ***percentage of AlpacaEval (correspondingly MT-Bench) samples on which Prometheus-v2.0 chose the refined response over the base policy's generation, with regards to the principle-following rubric.*** Recall that our STaR baseline also produces an intrinsic self-correction, but without the principle, so we use the principle invoked by the STaPLe model in the Prometheus judge rubric.

Table 3: Analysis of the Prometheus-8x7B-v2.0 model's judgements on the self-correction responses of the STaPLe model against the STaR baseline. The baseline win-rate against the base policy is 50%.

| Model | MT-Bench Prometheus Win-rate | AlpacaEval Prometheus Win-rate |
|---|---|---|
| **Llama-3.1-8B-Instruct** | | |
| STaR Iter 1 (28.2k) | 56.3% | 54.0% |
| STaPLe Iter 1 (28.2k) | 62.5% | 62.4% |
| STaR Iter 2 (6.0k) | 61.3% | 58.6% |
| STaPLe Iter 2 (6.1k) | 67.5% | 65.0% |
| STaR Iter 3 (6.1k) | 62.5% | 61.1% |
| STaPLe Iter 3 (6.3k) | **71.3%** | **68.7%** |
| STaR Iter 4 (6.3k) | 66.3% | 62.4% |
| STaPLe Iter 4 (6.6k) | 70.0% | 64.6% |
| **Granite-3.1-8B-Instruct** | | |
| STaR Iter 1 (24.1k) | 57.5% | 56.1% |
| STaPLe Iter 1 (24.1k) | 63.8% | 62.1% |
| STaR Iter 2 (5.4k) | 60.0% | 60.1% |
| STaPLe Iter 2 (5.2k) | 68.8% | 65.6% |
| STaR Iter 3 (5.9k) | 63.8% | 62.2% |
| STaPLe Iter 3 (5.9k) | 72.5% | **69.3%** |
| STaR Iter 4 (6.2k) | 66.3% | 63.0% |
| STaPLe Iter 4 (6.3k) | **73.8%** | 68.7% |
| **Qwen2.5-7B-Instruct** | | |
| STaR Iter 1 (30.9k) | 60.0% | 58.6% |
| STaPLe Iter 1 (30.9k) | 67.5% | 65.2% |
| STaR Iter 2 (6.5k) | 63.8% | 63.1% |
| STaPLe Iter 2 (6.5k) | 71.3% | 68.8% |
| STaR Iter 3 (7.1k) | 67.5% | 65.3% |
| STaPLe Iter 3 (7.0k) | 75.0% | 70.7% |
| STaR Iter 4 (7.1k) | 71.3% | 65.6% |
| STaPLe Iter 4 (7.1k) | **76.3%** | **71.3%** |

Both algorithms yield gains over the base policy in win-rate, with STaPLe outperforming STaR across all iterations. Interestingly, we find that on MT-Bench, the STaR baseline continues to increase by a sizable amount (2.5-3.8 pts) in iteration 4, unlike the true MT-Bench score and the other benchmarks as reported in Table 2. By contrast, training over principles in the unconstrained STaPLe yields a smaller gain (for Granite-8B and Qwen-7B, and a slight drop for Llama-8B), although STaPLe still outperforms STaR by +7.5-8.8% in iteration 3 and +3.7%-7.5% in iteration 4. However, Granite-8B does appear to improve in MT-Bench win-rate in iteration 4, despite the average MT-Bench score dropping (as can be witnessed in Table 2. However, given the small sample size of the dataset (80 samples), this could be a product of noise, unlike the larger datasets like IFEval (541 samples) and

AlpacaEval (805 samples). On AlpacaEval, we witness a similar trend, albeit more consistent with the AlpacaEval scores reported in Table 2.

## J  Stepwise Win-rate Analysis

Recall that the Prometheus win-rates that have been reported thus far are a comparison against generations from each model's initial policy (instruct model) $\pi_0$. However, to confirm that the model's generations continue to improve in principle-following quality over the iterations, we compare the iteration $t$ model's generations against iteration $t-1$ in the Prometheus judgement setup. Given our primary focus in Tables 1 and 2 was on IFEval, we recompute these win-rates against the initial response in trained STaPLe model's own generated self-correction trajectories. In iteration 1, the comparison is done against the base policy, and thus the win-rates reported are the same as in the aforementioned tables.

Table 4: Stepwise win-rates over the iterations of the unconstrained STaPLe algorithm with the Prometheus-v2.0 judge. Instead of comparing against the initial (instruction-tuned) policy for all iterations, this judge compares against the responses sampled from the previous iteration's policy.

| Model | IFEval Prometheus Win-rate |
|---|---|
| **Llama-3.1-8B-Instruct** | |
| STaPLe Iter 1 (28.2k) | 65.6% |
| STaPLe Iter 2 (6.1k) | 58.2% |
| STaPLe Iter 3 (6.3k) | 54.3% |
| STaPLe Iter 4 (6.6k) | 49.4% |
| **Granite-3.1-8B-Instruct** | |
| STaPLe Iter 1 (24.1k) | 65.1% |
| STaPLe Iter 2 (5.2k) | 62.3% |
| STaPLe Iter 3 (5.9k) | 58.0% |
| STaPLe Iter 4 (6.3k) | 47.9% |
| **Qwen2.5-7B-Instruct** | |
| STaPLe Iter 1 (30.9k) | 68.2% |
| STaPLe Iter 2 (6.5k) | 61.2% |
| STaPLe Iter 3 (7.0k) | 63.4% |
| STaPLe Iter 4 (7.1k) | 60.8% |

Note that a win-rate of 50% indicates that responses generated under $\pi_t$ were equally preferred to responses generated under $\pi_{t-1}$. As such, the win-rates remaining above 50% by a sizble margin is further evidence of the model's self-improvement. These stepwise win-rates are also a useful signal in behaving like an "elbow" method, to determine when to terminate the STaPLe algorithm. For instance, observing that the win-rates drop below 50% for the Llama-8B and Granite-8B models in iteration 4 suggests that their responses degraded compared to their prior iteration's responses (albeit, Llama-8B is fairly marginally below 50%). On the other hand, Qwen's win-rates remain above 60% throughout, suggesting that there perhaps is potential to continue its self-improvement for additional iterations. We plot this progression in Figure 4 for a visual representation of this selection process.

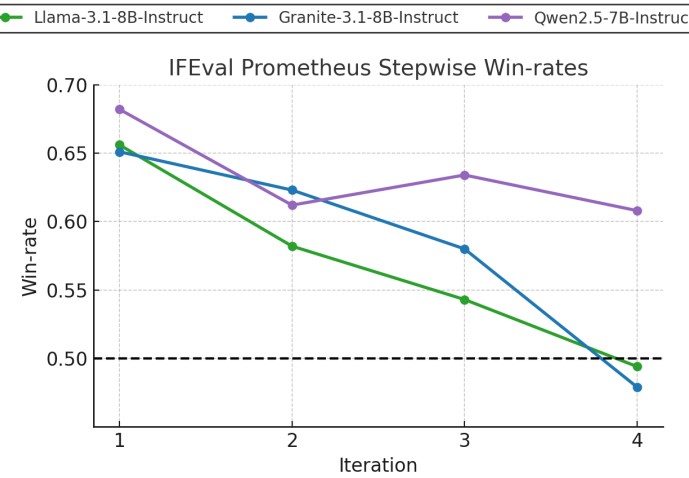

Figure 4: Visualization of Table 4, comparing against the 50% baseline. While the win-rate exceeds 50%, the model continues to self-improve.

## K   Intrinsic Self-Correction

Given that the trained STaPLe model performs intrinsic self-correction – given a prompt, it produces an initial response, invokes a principle to improve it, and improves the response, without an external stimulus or re-prompting – we can analyze the advantage between the model's initial and final responses. We do this using the Prometheus-v2.0 judge, on IFEval prompts, to give a binary preference between the initial response and final response on principle-following, using the same judge prompt as in other experiments in Tables 1-4. The results are found in Table 5. We find that the win-rates do improve over the iterations, reinforcing the claim that STaPLe-trained models learn intrinsic self-correction behavior. These win-rates are also consistent with our prior findings that the Llama-8B and Granite-8B models degrade in iteration 4, while Qwen-7B continues to improve.

Table 5: Prometheus-v2.0 win-rate in comparing the model-generated initial and refined responses, on the basis of which response better reflects the principle invoked for unconstrained STaPLe.

| Model | IFEval Prometheus Win-rate |
|---|---|
| **Llama-3.1-8B-Instruct** | |
| STaPLe Iter 1 (28.2k) | 72.6% |
| STaPLe Iter 2 (6.1k) | 74.3% |
| STaPLe Iter 3 (6.3k) | **75.0%** |
| STaPLe Iter 4 (6.6k) | 73.4% |
| **Granite-3.1-8B-Instruct** | |
| STaPLe Iter 1 (24.1k) | 76.5% |
| STaPLe Iter 2 (5.2k) | 77.1% |
| STaPLe Iter 3 (5.9k) | **83.2%** |
| STaPLe Iter 4 (6.3k) | 77.8% |
| **Qwen2.5-7B-Instruct** | |
| STaPLe Iter 1 (30.9k) | 75.8% |
| STaPLe Iter 2 (6.5k) | 78.0% |
| STaPLe Iter 3 (7.0k) | 79.7% |
| STaPLe Iter 4 (7.1k) | **82.1%** |

## K.1 Refinement Rate Analysis

Figure 5: STaPLe refinement rates across 4 iterations for unconstrained STaPLe algorithm. This represents the fraction of samples in the mining corpus on which at least one principle-conditioned refinement attempt improved over the initial response.

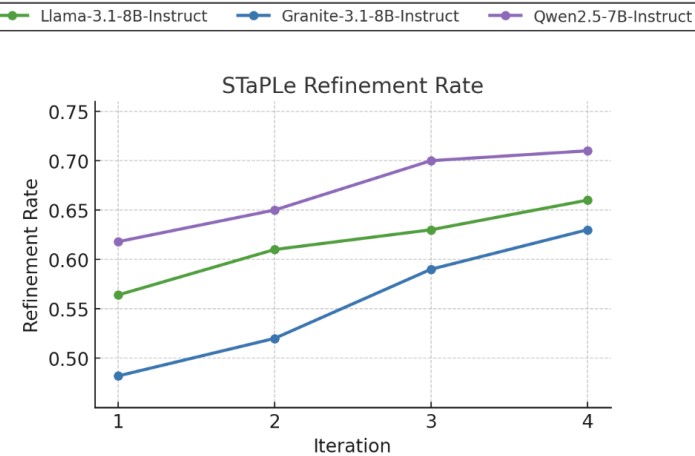

Figure 6: STaPLe refinement rates across 4 iterations for constrained STaPLe algorithm. This represents the fraction of samples in the mining corpus on which at least one principle-conditioned refinement attempt improved over the initial response.

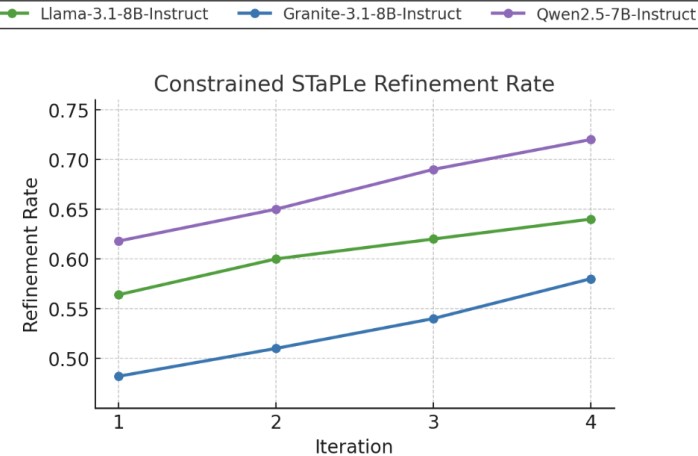

We compare the refinement rates between the unconstrained and constrained versions of STaPLe in Figures 5 and K.1. We observe a similar trend, where Qwen-7B starts with th highest rate (above 0.61) and remains the highest throughout. The refinement rates for Llama-8B and Granite-8B gain similarly for both versions, although the refinement rates are lower by iteration 4 in the constrained version. In the left plot, the Granite refinement rate spikes during iteration 3 principle discovery (the E-step), which we do not see in the constrained version.

## K.2 Precision Analysis in Self-Refinement

As weaker language models have been demonstrated to be less successful at inference-time self-refinement without targeted, pre-specified dimensions of refinement (Madaan et al., 2023; Ramji et al., 2024), we filter all refinements that did not improve over the initial response. To validate that although principles are proposed given $y^G$, the gold response itself is not "leaked" to the refinement trajectory, we compute the Rouge-L precision, and report the number of samples that exceed 0.9.

Table 6: Analyzing the number of samples with a high Rouge-L precision score.

| Model | Total Samples | # Samples with Rouge-L Precision > 0.9 | Percentage |
|---|---|---|---|
| **Llama-3.1-8B-Instruct** | | | |
| STaPLe Iter 1 | 28.2k | 91 | 0.32% |
| STaPLe Iter 2 | 6.1k | 35 | 0.57% |
| STaPLe Iter 3 | 6.3k | 31 | 0.49% |
| STaPLe Iter 4 | 6.6k | 41 | 0.62% |
| **Granite-3.1-8B-Instruct** | | | |
| STaPLe Iter 1 | 24.1k | 157 | 0.65% |
| STaPLe Iter 2 | 5.2k | 32 | 0.61% |
| STaPLe Iter 3 | 5.9k | 56 | 0.94% |
| STaPLe Iter 4 | 6.3k | 87 | 1.38% |
| **Qwen2.5-7B-Instruct** | | | |
| STaPLe Iter 1 | 30.9k | 176 | 0.57% |
| STaPLe Iter 2 | 6.5k | 53 | 0.82% |
| STaPLe Iter 3 | 7.0k | 88 | 1.25% |
| STaPLe Iter 4 | 7.1k | 82 | 1.16% |

The results can be found in Table 6 above; we find that the percentage of samples with a high precision does not exceed 1.5%, a negligibly small figure. Since the total number of samples (the denominator) consists of the instances where the refinement retained maximally improved in Rouge F1, if the precision is still relatively low, it must be that the recall improved. That is, while the rate of copying does not grow much, the generated responses better cover the gold – this is necessary for better reflecting the core elements of the gold response for the model to imitate with its on-policy generations. In works such as STaR (Zelikman et al., 2022), a chain-of-thought is induced that reconstructs the original gold answer — our work achieves a similar purpose, while instead learning a distribution of on-policy generations similar to the gold, rather than overfitting to the gold.

# L   Model-Generated Constitutions

For each model, we include the constitution generated, and a histogram of the densities of each element taught during the final iteration of training. This histogram denotes the number of samples in the cluster for which each principle serves as a representative.

## L.1   Granite-3.1-8B-Instruct-Generated Constitution

```
1.  Clarity and Conciseness
2.  Empathy and Compassion
3.  Contextualization
4.  Comprehensive Approach
5.  Balanced Information and Perspective
6.  Personalization and Empowerment
7.  Tone and Communication
8.  Ethical and Legal Considerations
9.  Precision and Specificity
10.  Accuracy and Verification
11.  Privacy and Consent
12.  Alternative Solutions
13.  Recipe Variety
14.  Detailed Support
```

```
15.  Sensory Imagery and Metaphors
16.  Step-by-Step Instructions
17.  Urgency and Action Steps
18.  Consequences and Prevention
19.  Holistic Approach
20.  Encouragement and Engagement
21.  Age and Developmental Appropriateness
22.  Incorporate More Narrative Elements
23.  Emphasize the importance of understanding the pet's natural habitat
and behavior
24.  Cultural Sensitivity
25.  Emphasize Objectivity and Respectfulness
26.  Emphasize Honesty and Integrity
27.  Emphasize Safety and Compliance
28.  Emphasize Individual Diversity and Avoid Stereotypes
29.  Consider Individual Preferences
30.  Emphasize the Importance of Professional Medical Advice
31.  Emphasize Practical Strategies
32.  Emphasize the Individual Nature of Relationships
33.  Emphasize the value of the individual and their experiences
34.  Simplification and Clarity
35.  Provide Specific Examples
```

Figure 7: Breakdown of the Granite-3.1-8B-Instruct iteration 3 model-generated constitution in terms of the number of elements in each cluster. The label on the x-axis denotes the cluster representative element (medoid). The counts also denote the number of fine-tuning samples contained this principle in the augmented dataset $\widetilde{\mathcal{D}}$, following label replacement in the trajectories. We use ellipses for the sake of readability.

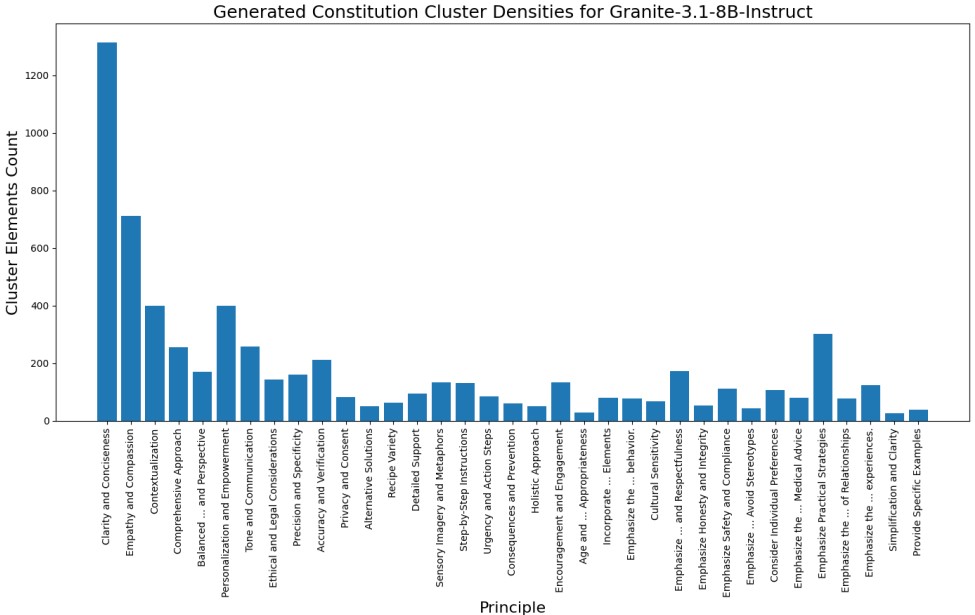

In particular, we observe that the "Clarity and Conciseness" and "Empathy and Compassion" principles are the most emphasized, likely as a result of mining corpus domains including summarization (TL;DR) and harmlessness (HH-RLHF). The phrase "Emphasize ..." is repeated fairly often, albeit in different contexts. This reflects the model's stylistic preferences for principles that aid it in self-correcting, one of the key reasons for using on-policy-generated principles in the STaPLe algorithm, rather than introducing "supervision" from a stronger model in an off-policy fashion. We also repeat

the Granite-generated constitution in Figure 7, to ease direct comparison of the constitutions across models here in the Appendix.

## L.2 Llama-3.1-8B-Instruct-Generated Constitution

```
1.  Directness and Assertiveness
2.  Conciseness and Clarity
3.  Empathy and Emotional Validation
4.  Structure and Organization
5.  Contextualization and Relevance
6.  Transparency and Honesty
7.  Avoiding Harm and Sensitivity
8.  Specificity and Completeness
9.  Avoiding Assumptions
10.  Challenge Assumptions
11.  Be More Direct and Clear in Addressing Concerns
12.  Acknowledge and Address Previous Misconceptions Clearly
13.  Simplicity and Gradual Complexity
14.  Be more assertive in responses to known information
15.  Simplify and Focus on the Relevant Information
16.  Redirect to a more acceptable alternative
17.  Avoid Technical Jargon and Focus on Clear Explanation
18.  Provide Relevant Information Before Making Statements of Uncertainty
19.  Empathize with the emotional state of the reader and explicitly
acknowledge their concerns before providing advice or solutions
20.  Avoiding Ambiguity through Clarity
21.  Empathic Validation and Clarification
22.  Precision Over Generality
23.  Minimize Unnecessary Information
24.  Provide a plausible yet incomplete answer or an educated guess if
direct information is not available
25.  Provide Clarifying Context and Additional Information When Necessary
26.  Engage in the conversation with a clear and well-defined tone
27.  Avoid providing unnecessary information
28.  Avoid unnecessary phrases and provide clear and direct information
29.  Address the reader's concerns explicitly
```

We observe that the at face value, the elements in the Llama-8B constituion are more "high-level", akin to some of the elements in works such as Constitutional AI and Dromedary (Bai et al., 2022b; Sun et al., 2023). As with Granite, the majority of the mass is placed on elements with the premise of "Conciseness and Clarity" (simply swapping the order), as well as "Empathy and Emotional Validation), which is fairly similar to "Empathy and Compassion" from the Granite-8B constitution. A new element that appears fairly often ($\approx 800$ instances) is "Directness and Assertiveness".

Figure 8: Analysis of the Llama-3.1-8B-Instruct iteration 4 constitution. We use ellipses for brevity, as in Figure 7, given the corresponding full principles may be found above.

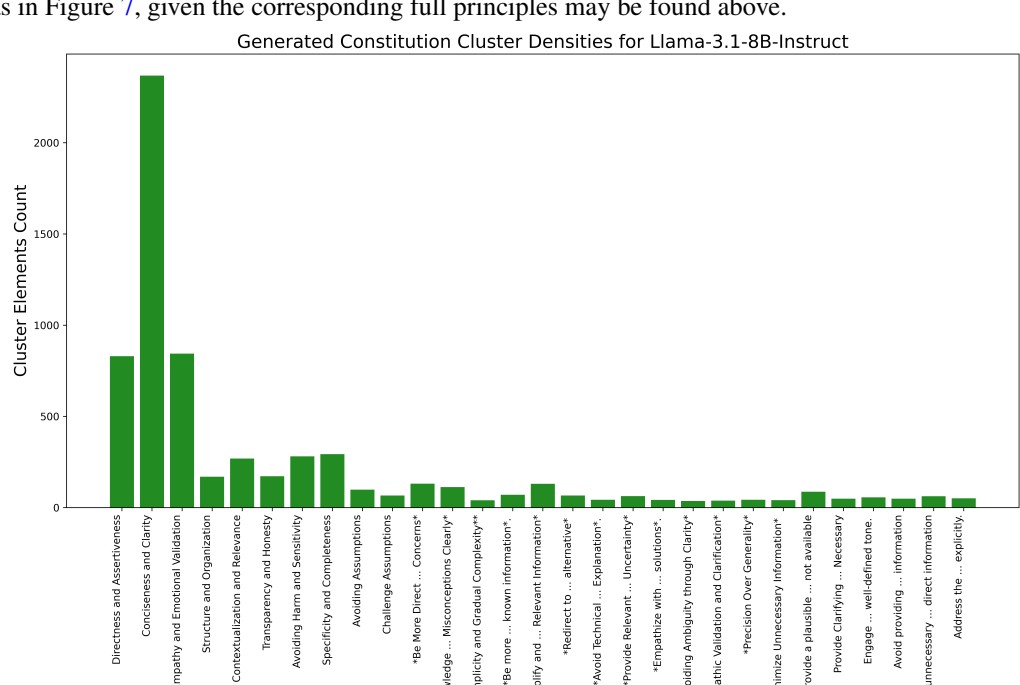

## L.3 Qwen2.5-7B-Instruct-Generated Constitution

```
1.   Respect for Privacy and Consent
2.   Clarity & Specificity
3.   Structured & Problem-Solving Guidance
4.   Comprehensive Consideration
5.   Inclusivity and Respect in Communication
6.   Specificity and Variety
7.   Empathy & Emotional Support
8.   Clarity of Information and Timeline Accuracy
9.   Clarify Assumptions and Context In Order to Provide Relevant
Information.
10.  Balanced Approach
11.  Consistency in Structure and Clarity
12.  Personal Growth Focused Guidance
13.  Comprehensive and Structured Guidance
14.  Ethics & Safety
15.  Emphasize Impact and Benefits.
16.  Clarity and Specificity in Response
17.  Consistent Structure and Flow
18.  Clarity and Consistency in Evidence Presentation
19.  Context & Background
20.  Respect and Communication
21.  Focus on Accuracy and Relevance To the Requested Information
22.  Clarify Distinctiveness of Breeds
23.  Contextual Humor and Tone Consistency
24.  Consistency in Genre Description
25.  Clear and Direct Answering
```

```
26.   Clarity and Specificity in Geographical Context
27.   Clarity and Structure in Explanations
28.   Clarify Ambiguity and Provide Specific Information
29.   Holistic Relationship Approach
30.   Emphasis on Comprehensive Support
31.   Clarity and Structure in Information Presentation
32.   Engagement and Specificity
33.   Consistent Narrative Elements
34.   Clarity and Specificity in Cultural Context
35.   Clarity and Contextual Relevance
36.   Clarity and Specificity in Context
37.   Clarity of Comparison
38.   Age-Appropriate Content and Supervision
39.   Consistency in Tone and Style
40.   Clarity and Specificity in Medical Conditions
41.   Personalization and Specificity
42.   Character Development and Consistency
43.   Clarity and Accuracy in Historical Context
44.   Clarity and Organization Through Structure
45.   Clarity and Specificity in Categorization
46.   Verify Information Accurately Before Providing
47.   Clarity and Structure in Information Provision
48.   Clarity and Relevance Principle
49.   Clarity and Focus on Key Information
```

Figure 9: Analysis of the Qwen2.5-7B-Instruct iteration 4 constitution. We use ellipses for brevity, as in Figures 7 and 8, given the corresponding full principles may be found above.

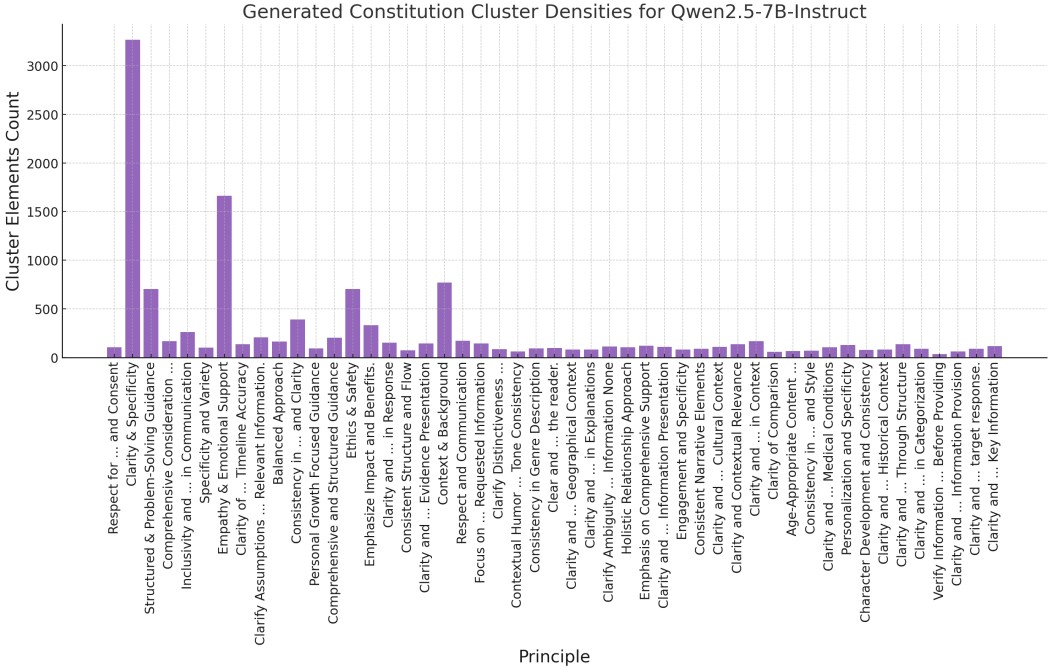

Qwen-7B appears to generate a larger constitution than the other models, despite discovering fewer new principles in subsequent iterations, as corroborated by Figure 3. However, we find the constitution to be, at face value, not as diverse in its phrasing given many of the principles have "clarity" or "clarify". However, the contexts behind its usage varies quite drastically, e.g. "Clarity of Information and Timeline Accuracy" differs greatly from "Clarity and Specificity in Cultural Context"; this is akin to the phrase "Emphasize" as noted earlier in the Granite constitutions. As such, we still find

this to be an appropriate constitution, especially when coupled with the gains that Qwen2.5-7B yields extending into the fourth iteration of STaPLe.

### L.4  Number of Clusters over the Iterations of STaPLe Algorithm

Figure 10: We plot the size of the constitutions generated under Constrained STaPLe with the medoids label replacement scheme.

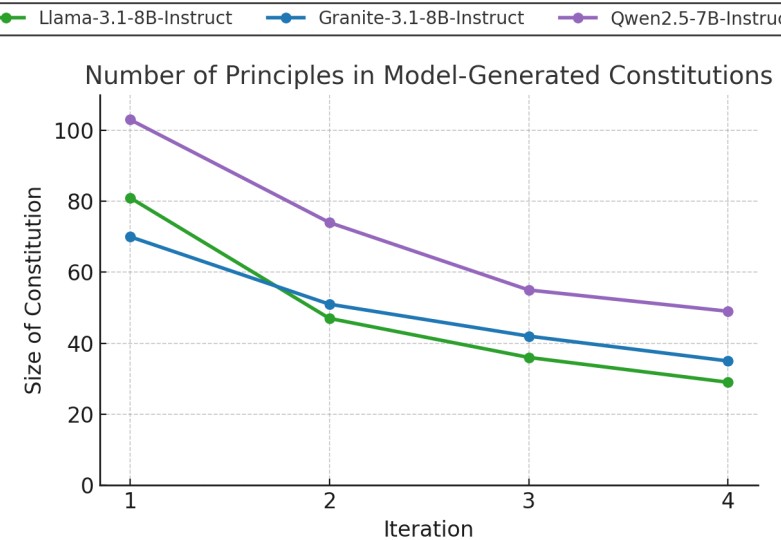

We observe that the size of the Qwen2.5-7B-generated constitution is larger throughout the iterations, although all models converge to a roughly fixed size, with the gap in size between the iterations 3 and 4 constitutions being minimal. The size of the constitution by iteration 4 is roughly around or more than 50% smaller than the iteration 1 constitution, suggesting that the learned distribution is converging to a stable set (surrounding this constitution). This also corroborates with Figure 3, where we show that the number of new principles discovered decreases over the iterations.

### L.5  Interpreting Model-Generated Constitutions

We include a breakdown of the constitution generated by Llama-3.1-8B-Instruct, as an example for how future taxonomy-oriented approaches might look to analyze the constitutions yielded by STaPLe. We categorize the principles into high-level domains: notably, "helpfulness", "relevance", "style", "safety", and "truthfulness and honesty", and find that the categories are nearly evenly balanced.

- Helpfulness: 6 principles (e.g. "Directness and Assertiveness")
- Style: 5 principles (e.g. "Structure and Organization")
- Relevance: 6 principles (e.g. "Minimize Unnecessary Information")
- Safety: 6 principles (e.g. "Avoiding Harm and Sensitivity")
- Truthfulness and Honesty: 6 principles ("Acknowledge and Address Previous Misconceptions Clearly")

## M  Inference-time Application of Discovered Constitutions

One natural question is: do the discovered constitutions hold practical utility *beyond interpretability*? We examine this by leveraging the discovered constitutions for *extrinsic* self-refinement, guided by prior work demonstrating that providing concrete refinement dimensions aids performance (Madaan et al., 2023; Ramji et al., 2024). However, rather than specifically selecting relevant principles for a domain a priori as in Ramji et al. (2024), we provide the complete constitution to the model in-context.

Given an initial generation and this constitution, we ask it to select a relevant principle to improve its generation, and to use it to perform refinement.

The results in Table 7 appear to follow a clear and encouraging trend: using better, on-policy-generated principles through STaPLe yields consistent gains over Self-Refine, but lags behind iterative training. Furthermore, the magnitude of gains is consistent with the strength of refinement capabilities observed prior: Llama-3.1-8B-Instruct is less proficient at self-refinement, so even providing seemingly better principles still yields modest returns. By contrast, Granite-3.1-8B-Instruct and Qwen2.5-7B-Instruct extract more substantial gains, concordant with Self-Refine outperforming the initial policy.

Table 7: Results of applying the final iteration constitutions along with the initial policy (the LM out-of-the-box) for *extrinsic*, rather than *intrinsic* self-refinement. STaPLe results reported below are the *unconstrained* version of the algorithm.

| Model | AlpacaEval |
|---|---|
| **Llama-3.1-8B-Instruct** | |
| Initial Policy | 26.9 |
| Self-Refine | 26.1 |
| Initial Policy + Iter 4 Constitution | 26.7 |
| STaPLe | 33.4 |
| **Granite-3.1-8B-Instruct** | |
| Initial Policy | 30.2 |
| Self-Refine | 31.7 |
| Initial Policy + Iter 4 Constitution | 33.1 |
| STaPLe | 38.4 |
| **Qwen2.5-7B-Instruct** | |
| Initial Policy | 30.4 |
| Self-Refine | 30.7 |
| Initial Policy + Iter 4 Constitution | 33.3 |
| STaPLe | 40.2 |

# N    Self-Improvement on Reasoning Models

While our procedure was designed primarily with boosting the instruction-following capabilities of weaker, small language models (SLMs), STaPLe can also be applied to stronger models, including those trained for advanced reasoning capabilities. We study the performance of Qwen3-32B with the "thinking" mode enabled, and compare STaPLe and STaR over three iterations, averaged over three experiments. Note that self-refinement is performed only over the final response, not over the thinking tokens (the <think>...</think> block). As exhibited by Table 8, STaPLe maintains a consistent gain over the iterations (+8.3%), and its margin of improvement compared to STaR is also relatively consistent, an encouraging result.

Table 8: Evaluating the performance of STaPLe and STaR with the Qwen3-32B reasoning model over 10k prompts per iteration.

| Model | AlpacaEval |
|---|---|
| **Qwen3-32B w/ Thinking Mode** | |
| Initial Policy | 60.5 |
| STaR Iter 1 (8.2k) | 63.0 |
| STaPLe Iter 1 (8.2k) | 65.5 |
| STaR Iter 2 (8.3k) | 64.7 |
| STaPLe Iter 2 (8.5k) | 67.2 |
| STaR Iter 3 (8.5k) | 66.3 |
| STaPLe Iter 3 (8.6k) | 68.6 |

# O STaPLe Algorithm Prompts

## O.1 Principle Mining Prompt

```
Prompt:  {prompt}

Here is the previous response:  {initial_response}

Here is the target response:  {gold_response}

Identify a high-level principle that may be useful to improve the quality
of this response to a human reader, to become more similar to the target
response.  If there is a principle that you can propose, provide it in
the format of 'New Principle: *[new principle name]*'.  Otherwise, if
there is no new principle that you can propose, respond with *[None]* at
the end of your response.
```

## O.2 Critique Generation Prompt

```
Prompt:  {prompt}
Response:  {curr_response}

Provide feedback on the above response, focusing entirely on how much
it addresses {principle}.  Be critical of the response, and how it can
improve relative to addressing {principle}

Feedback:
```

## O.3 Principle-Conditioned Refinement Prompt

```
Prompt:  {prompt}
Previous Response:  {curr_response}
Feedback:  {feedback}

Given this feedback on how the previous response addresses {principle},
improve the response on addressing {principle}"
Improved Response:
```

# P  Ablations

## P.1  Label Replacement Method

It is valuable to study *representativeness* during clustering, to ensure that the compressed set of principles reasonably retains the information contained in the complete set. This led us to a thorough investigation into the performance of the STaPLe algorithm under different label replacement methods. In particular, in addition to the medoid method outline in Section 3.1, we explore using the mode of each clustering based on the counts of principles invoked and an augmentation on the medoid scheme, where we only perform the label replacement if the difference in perplexity of the trajectory is bounded by a threshold $\tau_{PPL}$, which we take as 0.2.

$$\tilde{Z}_{medoid} = \{m_k : m_k = \arg \min_{m \in C_k} \sum_{j \in C_k} ||e_i - e_j||_2, \, k \in [1, K]\} \qquad \text{(Medoid Representatives)}$$

$$\tilde{Z}_{mode} = \{ m_k : m_k = \arg \max_{z \in C_k} \sum_{j \in C_k} \mathbf{1}(z_j = z), \quad k = 1, \ldots, K \} \qquad \text{(Mode Representatives)}$$

For the cluster medoid and mode label-replacement methods, we simply retrieve the cluster $C_i$ which sample $i$ belongs to, and replace $\hat{z}_i$ with $\tilde{z}_i$ from $\tilde{Z}_{medoid}$ or $\tilde{Z}_{mode}$, respectively. For the third method, define the perplexity of the sequence $S$ from the iteration $t$ language model $M_t$ to be $PPL(S; \theta_t) = \exp(-\frac{1}{|S|} \sum_{j=1}^{|S|} \ln[P_{\theta_t}(S_j \mid S_{<j})])$. We then compute the perplexity of the two sequence consisting of the input $x_i$, initial response $y_{i,1}$, principle candidate ($\hat{z}_i$ and $\tilde{z}_i$ from $\tilde{Z}_{medoid}$), critique based on the principle ($c_{\hat{z}_i}$ and $c_{\tilde{z}_i}$, respectively), and the refined response $y^2$ – denote these sequences $S_{i,\hat{z}_i}$ and $S_{i,\tilde{z}_i}$, respectively. If the difference in perplexity between these two sequences does not exceed a threshold $\tau$, we replace $\hat{z}_i$ with $\tilde{z}_i$ for $\widetilde{\mathcal{D}}$; else, we discard sample $i$. Intuitively, this means that all samples in $\widetilde{\mathcal{D}}$ with this perplexity difference scheme are those where the cluster medoid representative is nearly as good, if not better, than the original principle, based on likelihood of generation in the sequence, including the refined response. Formally, the set of principles retained are:

$$\tilde{Z}_{PPL} = \{ \tilde{z}_i \in \tilde{Z}_{medoid} \mid PPL(S_{i,\tilde{z}_i}; \theta_t) - PPL(S_{i,\hat{z}_i}; \theta_t) \leq \tau_{PPL} \}, i \in [1, |\mathcal{D}'|]\}$$

Regardless of the scheme, this results in dataset $(x_i, y_{i,1}, \tilde{z}_i, y_{i,2}) \in \widetilde{D}$, where $|\widetilde{\mathcal{D}}| \leq |\mathcal{D}'|$ for the perplexity method (equality otherwise).

The results of this analysis are included in Table 9. We find that using the medoid outperforms using the mode or the perplexity scheme (denoted PPL) across nearly all experiments, with the exception of Granite-8B iteration 4 for MT-Bench (average) and Qwen-7B in iteration 4 for AlpacaEval. That being said, the values across the schemes are generally close to one another, and follow a similar trend to the unconstrained version of STaPLe, suggesting that they are all viable principle cluster labels that may be taught to the LM. As noted in Section 4.2, STaPLe with clustering generally avoid the same degree of performance degradation seen in the unconstrained version for iteration 4 with Llama-8B and Granite-8B; this extends to the other two label replacement schemes as well. Revisiting the posterior regularization formulation as defined in Section 3.1, placing mass on a reduced number of elements induced by the clustering thus seems to, in fact, have a regularization effect of sorts.

Table 9: Comparison of the label replacement schemes proposed in Section 3.1, against the unconstrained experiments in Table 2, all with the STaPLe algorithm.

| Model | MT-Bench (avg) | MT-Bench (T1) | MT-Bench (T2) | AlpacaEval | IFEval WR |
|---|---|---|---|---|---|
| **Llama-3.1-8B-Instruct** | | | | | |
| Initial Policy | 7.46 | 8.09 | 6.83 | 26.9 | – |
| Unconstrained Iter 1 (28.2k) | 7.66 | 8.15 | 7.16 | 32.2 | 65.6% |
| Medoids Iter 1 (28.2k) | 7.63 | 8.14 | 7.11 | 31.9 | 65.1% |
| Modes Iter 1 (28.2k) | 7.59 | 8.10 | 7.09 | 31.2 | 64.5% |
| PPL Iter 1 (28.2k) | 7.62 | 8.14 | 7.09 | 31.1 | 64.3% |
| Unconstrained Iter 2 (6.1k) | **7.74** | **8.19** | 7.29 | 34.4 | 66.2% |
| Medoids Iter 2 (6.0k) | 7.70 | 8.15 | 7.25 | 34.6 | 66.0% |
| Modes Iter 2 (6.0k) | 7.66 | 8.14 | 7.18 | 33.8 | 65.1% |
| PPL Iter 2 (5.8kk) | 7.65 | 8.14 | 7.16 | 34.0 | 65.4% |
| Unconstrained Iter 3 (6.3k) | **7.74** | 8.16 | **7.31** | 35.6 | 68.8% |
| Medoids Iter 3 (6.2k) | 7.72 | 8.16 | 7.28 | **35.7** | 68.4% |
| Modes Iter 3 (6.2k) | 7.66 | 8.14 | 7.18 | 34.9 | 66.0% |
| PPL Iter 3 (6.1k) | 7.68 | 8.13 | 7.23 | 35.2 | 66.5% |
| Unconstrained Iter 4 (6.6k) | 7.71 | 8.13 | 7.30 | 33.4 | 68.9% |
| Medoids Iter 4 (6.4k) | 7.70 | 8.13 | 7.28 | 34.9 | **69.1%** |
| Modes Iter 4 (6.3k) | 7.63 | 8.13 | 7.14 | 34.1 | 66.7% |
| PPL Iter 4 (6.1k) | 7.68 | 8.11 | 7.25 | 33.7 | 66.7% |
| **Granite-3.1-8B-Instruct** | | | | | |
| Initial Policy | 7.83 | 8.59 | 7.08 | 30.2 | – |
| Unconstrained Iter 1 (24.1k) | 7.99 | 8.69 | 7.29 | 36.7 | 65.1% |
| Medoids Iter 1 (24.1k) | 7.98 | 8.66 | 7.30 | 36.2 | 64.9% |
| Modes Iter 1 (24.1k) | 7.94 | 8.69 | 7.19 | 35.8 | 64.0% |
| PPL Iter 1 (24.1k) | 7.93 | 8.64 | 7.23 | 35.2 | 63.3% |
| Unconstrained Iter 2 (5.2k) | 8.04 | 8.74 | 7.34 | 38.9 | 65.2% |
| Medoids Iter 2 (5.1k) | 8.01 | 8.68 | 7.35 | 38.7 | 67.3% |
| Modes Iter 2 (5.1k) | 7.98 | 8.71 | 7.25 | 37.8 | 65.6% |
| PPL Iter 2 (4.8k) | 7.99 | 8.65 | 7.33 | 38.1 | 66.7% |
| Unconstrained Iter 3 (5.9k) | **8.06** | **8.75** | 7.38 | **39.8** | **71.6%** |
| Medoids Iter 3 (5.4k) | **8.06** | 8.74 | 7.39 | 39.4 | 69.9% |
| Modes Iter 3 (5.3k) | 8.02 | 8.74 | 7.30 | 38.9 | 68.0% |
| PPL Iter 3 (5.2k) | 8.05 | 8.73 | 7.38 | 39.1 | 68.6% |
| Unconstrained Iter 4 (6.3k) | 8.04 | 8.66 | **7.41** | 38.4 | 67.6% |
| Medoids Iter 4 (5.8k) | 8.03 | 8.65 | **7.41** | 38.8 | 68.4% |
| Modes Iter 4 (5.5k) | 8.01 | 8.68 | 7.35 | 37.3 | 67.1% |
| PPL Iter 4 (5.3k) | 8.04 | 8.65 | 7.43 | 38.2 | 67.7% |
| **Qwen2.5-7B-Instruct** | | | | | |
| Initial Policy | 6.83 | 7.34 | 6.31 | 30.4 | – |
| Unconstrained Iter 1 (30.9k) | 7.03 | 7.48 | 6.59 | 37.3 | 68.2% |
| Medoids Iter 1 (30.9k) | 6.99 | 7.43 | 6.55 | 36.5 | 67.3% |
| Modes Iter 1 (30.9k) | 6.97 | 7.43 | 6.51 | 36.3 | 67.3% |
| PPL Iter 1 (30.9k) | 6.97 | 7.40 | 6.54 | 36.5 | 66.9% |
| Unconstrained Iter 2 (6.5k) | 7.14 | 7.55 | 6.73 | 39.4 | 66.2% |
| Medoids Iter 2 (6.5k) | 7.10 | 7.46 | 6.74 | 38.9 | 68.4% |
| Modes Iter 2 (6.5k) | 7.08 | 7.48 | 6.68 | 38.5 | 67.3% |
| PPL Iter 2 (6.3k) | 7.09 | 7.46 | 6.73 | 38.5 | 67.7% |
| Unconstrained Iter 3 (7.0k) | 7.20 | 7.63 | 6.78 | 39.8 | 72.5% |
| Medoids Iter 3 (6.9k) | 7.17 | 7.54 | 6.80 | 39.8 | 70.4% |
| Modes Iter 3 (6.9k) | 7.12 | 7.54 | 6.70 | 39.2 | 68.8% |
| PPL Iter 3 (6.8k) | 7.15 | 7.53 | 6.78 | 39.6 | 69.7% |
| Unconstrained Iter 4 (7.1k) | **7.24** | **7.64** | **6.85** | **40.2** | **73.4%** |
| Medoids Iter 4 (7.2k) | 7.22 | 7.60 | 6.84 | 39.9 | 72.1% |
| Modes Iter 4 (7.1k) | 7.14 | 7.56 | 6.73 | 39.1 | 69.7% |
| PPL Iter 4 (7.1k) | 7.17 | 7.55 | 6.79 | 40.0 | 71.0% |

## P.2 LLM-as-a-Judge Rejection Sampling

We note in Section 3 and 4.1 that we use the Rouge-L F1 score as the similarity scoring metric between a candidate response and the gold reference. We find this method to work well in practice, as shown by the results thus far. Nonetheless, under the recent paradigm of using an LLM-as-a-judge (Zheng et al., 2023), one could use a stronger performing model as a judge to score closeness to the gold, provided that one is willing to expend the inference-time compute to do so. We explore this setup using the Phi-4 model (Abdin et al., 2024), a 14B parameter model which reduces latency in performing $N + 1$ judge queries (one per refined response, along with the initial response), compared to a larger model such as Mixtral-8x22B or Llama-3.1-405B-Instruct. We use a score threshold of 9 on a scale from 1-10 for the initial response – that is, if the model assigns a score of 8 or lower, we proceed to refinement.

### P.2.1 Judge Prompt for Similarity Scoring

We leverage a judge prompt adapted from Katsis et al. (2025), focusing on comparison against the reference answer rather than faithfulness to a grounding document.

```
## System
[Instruction]
Please act as an impartial judge and evaluate the quality of the response
provided by an AI assistant to the user question given the provided
document and a reference answer."

## User
Your evaluation should assess the faithfulness, appropriateness, and
completeness.  Your evaluation should focus on the assistant's answer to
the question of the current turn.  You will be given the assistant's
answer and a sample reference answer.  You will also be given the
user questions and assistant's answers of the previous turns of the
conversation.  You should consider how well the assistant's answer
captures the key information, knowledge points mentioned in the reference
answer, when appropriate, and how it respects or builds upon the focus
and knowledge points from the previous turns.

[Appropriateness]:  You should evaluate if the assistant's answer is
relevant to the question of the current turn and if it addresses all the
issues raised by the question without adding extra information.

[Completeness]:  You should evaluate whether the assistant's answer is
complete with information from the reference.  Begin your evaluation by
comparing the assistant's answer against the reference answer in this
turn.  Be as objective as possible, and provide a detailed justification
for your rating.  You must rate the response on a scale of 1 to 10 and
providing a justification.  Return your response in the following format:
{"score":  your_score, "justification":  your_justification}

[INPUT]
{prompt}

[REFERENCE]
{gold}

[PREDICTION]
{response}
```

Table 10: Self-improvement with the Constrained STaPLe algorithm using a Phi-4 model as a judge to score similarity to the gold response for rejection sampling. We include Constrained STaPLe with Rouge-L, to make a direct comparison, denoted "STaPLe w/ Rouge".

| Model | MT-Bench (avg) | MT-Bench (T1) | MT-Bench (T2) | AlpacaEval | IFEval WR |
|---|---|---|---|---|---|
| **Llama-3.1-8B-Instruct** | | | | | |
| Initial Policy | 7.46 | 8.09 | 6.83 | 26.9 | – |
| STaR Iter 1 (28.2k) | 7.43 | 8.04 | 6.81 | 29.1 | 55.5% |
| STaPLe w/ Rouge Iter 1 (28.2k) | 7.63 | 8.14 | 7.11 | 31.9 | 65.1% |
| STaPLe w/ Judge Iter 1 (25.8k) | 7.60 | 8.13 | 8.08 | 31.6 | 64.9% |
| STaR Iter 2 (6.0k) | 7.47 | 8.08 | 6.86 | 30.6 | 57.7% |
| STaPLe w/ Rouge Iter 2 (6.0k) | 7.70 | 8.15 | 7.25 | 34.6 | 66.0% |
| STaPLe w/ Judge Iter 2 (5.7k) | 7.68 | 8.15 | 7.21 | 34.1 | 65.6% |
| STaR Iter 3 (6.1k) | 7.51 | 8.10 | 6.91 | 31.5 | 61.0% |
| STaPLe w/ Rouge Iter 3 (6.2k) | **7.72** | **8.16** | **7.28** | **35.7** | **68.4%** |
| STaPLe w/ Judge Iter 3 (6.3k) | 7.70 | **8.16** | 7.25 | 35.6 | 68.0% |
| **Granite-3.1-8B-Instruct** | | | | | |
| Initial Policy | 7.83 | 8.59 | 7.08 | 30.2 | – |
| STaR Iter 1 (24.1k) | 7.83 | 8.61 | 7.05 | 33.0 | 57.3% |
| STaPLe w/ Rouge Iter 1 (24.1k) | 7.98 | 8.66 | 7.30 | 36.2 | 64.9% |
| STaPLe w/ Judge Iter 1 (20.9k) | 7.93 | 8.66 | 7.20 | 36.0 | 65.2% |
| STaR Iter 2 (5.4k) | 7.86 | 8.63 | 7.10 | 34.7 | 59.5% |
| STaPLe w/ Rouge Iter 2 (5.1k) | 8.01 | 8.68 | 7.35 | 38.7 | 67.3% |
| STaPLe w/ Judge Iter 2 (5.2k) | 8.01 | 8.70 | 7.31 | 39.0 | 66.9% |
| STaR Iter 3 (5.9k) | 7.92 | 8.66 | 7.18 | 35.4 | 61.9% |
| STaPLe w/ Rouge Iter 3 (5.4k) | 8.06 | 8.74 | **7.39** | 39.4 | 69.9% |
| STaPLe w/ Judge Iter 3 (6.3k) | **8.07** | **8.76** | 7.38 | **40.4** | **70.2%** |
| **Qwen2.5-7B-Instruct** | | | | | |
| Initial Policy | 6.83 | 7.34 | 6.31 | 30.4 | – |
| STaR Iter 1 (30.9k) | 6.85 | 7.39 | 6.31 | 34.5 | 61.0% |
| STaPLe w/ Rouge Iter 1 (30.9k) | 6.99 | 7.43 | 6.55 | 36.5 | 67.3% |
| STaPLe w/ Judge Iter 1 (29.5k) | 6.96 | 7.45 | 6.48 | 36.2 | 66.7% |
| STaR Iter 2 (6.5k) | 6.98 | 7.45 | 6.51 | 36.9 | 63.0% |
| STaPLe w/ Rouge Iter 2 (6.5k) | 7.10 | 7.46 | 6.74 | 38.9 | 68.4% |
| STaPLe w/ Judge Iter 2 (6.5k) | 7.05 | 7.48 | 6.63 | 38.1 | 67.7% |
| STaR Iter 3 (7.1k) | 7.08 | 7.58 | 6.59 | 37.6 | 66.4% |
| STaPLe w/ Rouge Iter 3 (6.9k) | **7.17** | 7.54 | **6.80** | **39.8** | **70.4%** |
| STaPLe w/ Judge Iter 3 (7.2k) | 7.13 | **7.56** | 6.70 | 39.5 | 69.5% |

### P.2.2 Results

In Table 11, we present a similar table as Table 2, but comparing STaPLe over 3 iterations with the judge for rejection sampling in place of Rouge-L scoring. We use constrained STaPLe with the medoids label replacement method. We observe that the MT-Bench average scores drop slightly relative to using the Rouge-L similarity function, but still vastly outperforming STaR; in fact for Granite-8B, the iteration 2 scores are equal and STaPLe with the Phi-4 judge actually outperforms it in iteration 3. Notably, the turn-1 scores are higher with the Phi-4 while the turn-2 scores drop. On AlpacaEval, the scores of STaPLe with the judge are slightly lower than with Rouge-L for Llama-8B and Qwen-7B, while they gain +1% in iteration 3 for Granite-8B. A similar trend persists for the IFEval Win-rates, where Granite gains slightly in iterations 1 and 3, while Qwen and Llama drop slightly. We conclude that given the scores are largely similar, this highlights the generality of the STaPLe algorithm in expanding to various choices of similarity function.

## P.3 Bayesian Hyperparameter Optimization for Clustering Distance Threshold

As discussed in Section 4.1, we use the deterministic agglomerative clustering algorithm to ensure a fast, yet consistent assignment of clusters over the principle embeddings. However, this relies on a hyperparameter, $\delta$, which we use to denote the Euclidean distance threshold under which clusters will be merged. As such, a lower threshold corresponds to a greater number of clusters, and vice versa, thus controlling the size of the yielded constitutions. This hyperparameter is currently set in a manual fashion, where the size and representative elements (medoids or modes) are inspected by the authors of this work and the threshold adjusted if needed – this resulted in thresholds of 6 (Granite) and 8 (Llama and Qwen) for iteration 1, which was subsequently decreased to 5 and 7, respectively, for iterations 2-4. However, it is desirable for this threshold to be adaptive, and to mathematically encode the target properties for a cluster to satisfy.

Accordingly, we design an objective function consisting of two terms over a clustering assignment: 1. the inter-medoid diversity and 2. the intra-cluster tightness. The former is denoted by the average cosine-similarity (abbreviated as "cossim" henceforth) between each pair of medoids, while the latter is average cosine similarity of the points in their cluster to their own medoid. This can be written mathematically as follows:

$$
J(\delta) = \lambda \cdot \frac{2}{|C|(|C|-1)} \sum_{1 \leq i < j \leq |C|} [1 - cossim(m_i, m_j)] + (1-\lambda) \cdot \frac{1}{|C|} \sum_{k=1}^{|C|} \frac{1}{|C_k|} \sum_{i \in C_k} cossim(z_i, m_k)
$$

where $C = AggClustering(\delta)$ is the set of clusters assigned by the Agglomerative Clustering algorithm at a threshold of $\delta$. Given we value a balance between medoid diversity and intra-cluster tightness to the medoid for higher quality assignments, we set $\lambda = 0.5$ to weigh both terms equally.

We aim to search for a value of $\delta$ that yields a clustering $C$ that maximizes this objective. We use the scikit-optimize package (Head et al., 2020) to perform Bayesian optimization via Gaussian Processes to search for an optimal value of $\delta$ over this function. This process performs Gaussian Process regression over seen instances, uses an expected improvement $(-\mathbb{E}[J(x) - J(x^+)]$ acquisition function to identify the next threshold to evaluate, then clusters and evaluates at the next chosen value of $x$, repeating this process iteratively. We use the L-BFGS algorithm over 30 evaluations.

We evaluate the STaPLe algorithm with the Llama-8B and Granite-8B models at the chosen thresholds $\delta_i^*$ for iteration $i$, which we also report below. Following from the results in Table 9, where the medoid label replacement scheme performs the best, we apply this to the clusters yielded using $\delta_i^*$.

Notably, it is interesting that the thresholds drop in iteration 3, suggesting that as the number of clusters decreases, a more permissive threshold suffices to balance diversity and cluster tightness. We find that the optimized thresholds result in very similar results, albeit with slight improvements across both MT-Bench turns, and thus the average score as well.

### P.3.1 Diversity of Constitutions with Manually Selected Thresholds

To further study the claim of the original, hand-set thresholds being fairly well optimized, we can use this this objective function $J(\delta)$ as an appropriate metric to study the quality of the clusterings yielded. As expected, the optimized thresholds improve diversity, by a sufficient margin to suggest that there exist multiple thresholds which would improve upon the manually set $\delta_i$ values; the best of which being these $\delta_i^*$ values.

# Q   Qualitative Examples of Principle-Guided Self-Correction

## Q.1   Llama-3.1-8B-Instruct IFEval Examples

Note that the "Initial Response:" tags in the examples below are added for illustrative purposes, and are *not* generated at inference-time; the other tags "Principle:" and "Refined Response:" *are* generated to clearly indicate self-correction to the user.

Table 11: Analyzing the Constrained STaPLe algorithm performance over three iterations with optimal thresholds on the diversity-tightness objective searched over via Bayesian hyperparameter optimization. We use the medoids label replacement scheme, among the options in Appendix P.1.

| Model | MT-Bench (avg) | MT-Bench (T1) | MT-Bench (T2) | AlpacaEval |
|---|---|---|---|---|
| **Llama-3.1-8B-Instruct** | | | | |
| Initial Policy | 7.46 | 8.09 | 6.83 | 26.9 |
| STaR Iter 1 (28.2k) | 7.43 | 8.04 | 6.81 | 29.1 |
| STaPLe ($\delta_1 = 8.0$) Iter 1 (28.2k) | 7.63 | 8.14 | 7.11 | 31.9 |
| STaPLe ($\delta_1^* = 7.2$) Iter 1 (28.2k) | 7.64 | 8.14 | 7.14 | 31.8 |
| STaR Iter 2 (6.0k) | 7.47 | 8.08 | 6.86 | 30.6 |
| STaPLe ($\delta_2 = 7.0$) Iter 2 (6.0k) | 7.70 | 8.15 | 7.25 | 34.6 |
| STaPLe ($\delta_2^* = 7.3$) Iter 2 (6.1k) | 7.72 | 8.16 | 7.28 | 34.6 |
| STaR Iter 3 (6.1k) | 7.51 | 8.10 | 6.91 | 31.5 |
| STaPLe ($\delta_3 = 7.0$) Iter 3 (6.2k) | 7.72 | 8.16 | 7.28 | **35.7** |
| STaPLe ($\delta_3^* = 6.6$) Iter 3 (6.4k) | **7.75** | **8.18** | **7.33** | 35.4 |
| **Granite-3.1-8B-Instruct** | | | | |
| Initial Policy | 7.83 | 8.59 | 7.08 | 30.2 |
| STaR Iter 1 (24.1k) | 7.83 | 8.61 | 7.05 | 33.0 |
| STaPLe ($\delta_1 = 6.0$) Iter 1 (24.1k) | 7.98 | 8.66 | 7.30 | 36.2 |
| STaPLe ($\delta_1^* = 6.3$) Iter 1 (24.1k) | 7.98 | 8.65 | 7.31 | 36.0 |
| STaR Iter 2 (5.4k) | 7.86 | 8.63 | 7.10 | 34.7 |
| STaPLe ($\delta_2 = 5.0$) Iter 2 (5.1k) | 8.01 | 8.68 | 7.35 | 38.7 |
| STaPLe ($\delta_2^* = 5.9$) Iter 2 (5.2k) | 8.02 | 8.68 | 7.36 | 38.8 |
| STaR Iter 3 (5.9k) | 7.92 | 8.66 | 7.18 | 35.4 |
| STaPLe ($\delta_3 = 5.0$) Iter 3 (5.4k) | 8.06 | 8.74 | 7.39 | **39.4** |
| STaPLe ($\delta_3^* = 4.2$) Iter 3 (5.9k) | **8.08** | **8.75** | **7.41** | 39.2 |

### Q.1.1 Principle: Conciseness and Clarity

Prompt: Write a short blog post about a trip to Japan using less than 300 words.

Initial Response: Japan is a country that never fails to amaze. From the bustling streets of Tokyo to the serene temples of Kyoto, every corner of this island nation offers a unique experience.

Our journey began in Tokyo, a city that seamlessly blends the traditional and the modern. We marveled at the neon-lit skyscrapers of Shibuya, watched the world's busiest pedestrian crossing, and indulged in sushi at Tsukiji Fish Market. The city's vibrant energy was infectious, and we found ourselves captivated by its charm.

Next, we headed to Kyoto, a city that feels like stepping back in time. The ancient temples, like Kinkaku-ji and Fushimi Inari, were a testament to Japan's rich history. The tranquil gardens and the ritual of tea ceremony were a stark contrast to the city's hustle and bustle.

No trip to Japan would be complete without experiencing its natural beauty. We ventured to Hakone, a mountainous region known for its hot springs and stunning views of Mt. Fuji. The scenic train ride through the Japanese Alps was a highlight of our trip.

Japan's culinary scene is as diverse as its landscapes. From the humble bento box to the exquisite kaiseki, every meal was a gastronomic delight. The attention to detail and the emphasis on seasonal ingredients were evident in every dish.

Our trip to Japan was more than just a vacation; it was a cultural immersion. The country's blend of tradition and modernity, its respect for nature, and its culinary artistry left an indelible mark on

Table 12: Analyzing the constitutions yielded by the Constrained STaPLe algorithm, both with and without the Bayesian hyperparameter optimization process detailed above.

| Model | $J(\delta)$ |
|---|---|
| **Llama-3.1-8B-Instruct** | |
| STaPLe ($\delta_1 = 8.0$) Iter 1 (28.2k) | 0.6437 |
| STaPLe ($\delta_1^* = 7.6$) Iter 1 (28.2k) | 0.6502 |
| STaPLe ($\delta_2 = 7.0$) Iter 2 (6.0k) | 0.6625 |
| STaPLe ($\delta_2^* = 7.3$) Iter 2 (6.1k) | 0.6732 |
| STaPLe ($\delta_3 = 7.0$) Iter 3 (6.2k) | 0.6889 |
| STaPLe ($\delta_3^* = 6.6$) Iter 3 (6.4k) | 0.7054 |
| **Granite-3.1-8B-Instruct** | |
| STaPLe ($\delta_1 = 6.0$) Iter 1 (24.1k) | 0.6036 |
| STaPLe ($\delta_1^* = 6.3$) Iter 1 (24.1k) | 0.6151 |
| STaPLe ($\delta_2 = 5.0$) Iter 2 (5.1k) | 0.6241 |
| STaPLe ($\delta_2^* = 5.9$) Iter 2 (5.2k) | 0.6482 |
| STaPLe ($\delta_3 = 5.0$) Iter 3 (5.4k) | 0.6765 |
| STaPLe ($\delta_3^* = 4.2$) Iter 3 (5.9k) | 0.6894 |

us. Japan, with its myriad charms, is a place that will forever hold a special place in our hearts.

In less than 300 words, we've tried to encapsulate the essence of Japan - a country that is as complex as it is captivating. From its bustling cities to its serene temples, from its technological marvels to its natural wonders, Japan is a land of contrasts that offers something for every traveler.

**Principle: Conciseness and Clarity**

Refined Response: Japan, a land of contrasts, captivated us with its blend of tradition and modernity. Our journey began in Tokyo, where neon-lit skyscrapers and bustling streets of Shibuya contrasted with the serenity of Kinkaku-ji, or the Golden Pavilion. We savored sushi at Tsukiji Fish Market, a testament to Japan's culinary artistry.

Next, Kyoto's ancient temples and tranquil gardens offered a stark contrast to Tokyo's vibrant energy. The Fushimi Inari shrine, with its thousands of vermilion torii gates, was a spiritual highlight.

In Hakone, we experienced Japan's natural beauty. The scenic train ride through the Japanese Alps, coupled with the soothing hot springs, provided a serene escape.

Japan's culinary scene, from humble bento boxes to exquisite kaiseki, showcased the country's respect for seasonal ingredients and meticulous attention to detail.

Our trip to Japan was a cultural immersion, leaving an indelible mark. The country's contrasts - from bustling cities to serene temples, technological marvels to natural wonders - make it a must-visit destination for every traveler.

While the initial response is a reasonably well-written blog about Japan, it exceeds the 300 word limit, thus failing to follow the instruction. The trained STaPLe model recognizes this intrinsically, invokes the "Conciseness and Clarity" principle by observing that concision is the primary lacking element of the initial response, and then produces a refined response. The resulting response falls well under the 300 word limit, while paraphrasing much of the prior blog post's core elements, thus accomplishing the task.

### Q.1.2 Principle: Specificity and Completeness

> Prompt: I'm interested in a college with open enrollment and a regional accreditation. Which college would you recommend? Don't include the keywords "DuPage" and "Dade" in your response. Let's make it a constrained writing problem: be sure the letter p appears at least 15 times in your response.
>
> Initial Response: I'd recommend considering the University of the People (UOP).
>
> **Principle: Specificity and Completeness**
>
> Refined Response: I'd recommend considering the University of the People (UOP), a non-profit institution with open enrollment and regional accreditation by the Higher Learning Commission (HLC). UOP offers affordable, accredited bachelor's and master's degrees in business administration, computer science, and health science. Their tuition-free model and flexible schedule make it an excellent choice for non-traditional students.
>
> UOP's open enrollment policy means that anyone with a high school diploma or equivalent can apply, regardless of previous academic experience. This inclusivity aligns with your request for a college with open enrollment.
>
> The university's regional accreditation by the HLC ensures that its degrees are recognized and respected by employers, academic institutions, and professional organizations. This accreditation is crucial for maintaining the value and transferability of your degree.
>
> UOP's online, asynchronous learning format allows students to balance their education with work and personal commitments. This flexibility is particularly beneficial for adult learners and those in remote areas.
>
> Moreover, UOP's commitment to affordability is commendable. As a non-profit, the university offers tuition-free degrees, making higher education accessible to a broader range of students.
>
> In summary, the University of the People is a college with open enrollment and regional accreditation that meets your criteria. Its flexible, affordable, and accredited programs make it an excellent choice for students seeking a high-quality education without the burden of high tuition fees.

The initial response produces the name of a college, but does not provide a particularly informative response. It also does not meet the criterion of including the letter "p" at least 15 times in its response. The model intrinsically recognizes the need for greater specificity in its answer, and produces a more substantial and useful revised response.

### Q.1.3 Principle: Directness and Assertiveness

> Prompt: Write a cover letter for a job and end with exactly "Call me at 631-481-4867" No other words should follow that phrase.
>
> Initial Response:
> Dear Hiring Manager,
>
> I am writing to express my interest in the [Job Title] position at [Company Name], as advertised. With a Bachelor's degree in [Relevant Field] and over [Number of Years] years of experience in [Relevant Skills], I am confident that I would be a valuable addition to your team.
>
> In my previous role at [Previous Company Name], I was responsible for [Key Responsibility 1], which allowed me to hone my skills in [Skill 1]. I also led a team of [Number of Team Members] in [Project/Initiative], resulting in [Achievement]. I am particularly drawn to [Company Name] because of its reputation for [Company Value/Mission], which aligns with my own professional

goals.

I am excited about the opportunity to bring my unique blend of skills and experience to [Company Name] and am eager to contribute to your team's success. I am available at your earliest convenience for an interview and can be reached at 631-481-4867.

Thank you for considering my application. I look forward to the opportunity to discuss how I can contribute to your team.

Sincerely,
[Your Name]

Call me at 631-481-4867.

**Principle: Directness and Assertiveness**

Refined Response: Dear Hiring Manager,

I am excited to apply for the [Job Title] position at [Company Name]. With a Bachelor's in [Relevant Field] and [Number of Years] years of experience in [Relevant Skills], I am confident in my ability to excel in this role.

In my previous role at [Previous Company Name], I honed my skills in [Skill 1] through [Key Responsibility 1], and led a team of [Number of Team Members] to achieve [Achievement]. I am drawn to [Company Name] because of its commitment to [Company Value/Mission], which resonates with my professional aspirations.

I am eager to bring my unique blend of skills and experience to [Company Name] and contribute to your team's success. I am available for an interview at your earliest convenience and can be reached at 631-481-4867.

Thank you for considering my application. I look forward to discussing how I can contribute to your team.

Sincerely, [Your Name]

Call me at 631-481-4867.

This minor rewrite reflects a more assertive tone, as indicated in the principle invoked by the model, "Directness and Assertiveness". Specifically, when comparing phrasing such as "discussing how I can contribute ..." as opposed to "the opportunity to discuss ...", the former is evidently more direct and confident in pursuing this job opportunity.

### Q.2 Granite-3.1-8B-Instruct IFEval Examples

### Q.2.1 Principle: "Urgency and Action Steps"

Prompt: Please rewrite the following sentence in a serious tone, similar to the president of the united states, and without an upward bias or inflation alarmism. Please also include the keywords "ink" and "memoirs" in the sentence: "The main points to consider are the following: First, the ECB's own forecasts, which currently have a strong upward bias, must be revised downwards. Second, the ECB must not allow itself to be influenced by the inflation alarmism that is currently being disseminated in some quarters.

Initial Response: In the context of our ongoing analysis, it is imperative to acknowledge the significance of the European Central Bank's projections, which have been marked by a pronounced upward trend. It is crucial that these forecasts are recalibrated to reflect a more balanced perspective. Furthermore, it is essential that the ECB remains steadfast in its

decision-making process, unswayed by the inflation alarmism that has been circulating in certain circles.

**Principle: Urgency and Action Steps**

Refined Response: In the course of our meticulous examination, it is of paramount importance to recognize the profound implications of the European Central Bank's projections, which have been characterized by a marked upward trajectory. It is incumbent upon us to adjust these forecasts to better reflect a more nuanced and balanced outlook. Moreover, it is of critical significance that the ECB maintains its resolute stance in the face of the inflation alarmism that has been pervasive in certain sectors. This is a matter of utmost urgency, as we endeavor to navigate the intricate complexities of our economic terrain. In this regard, I have recently put ink to paper in my memoirs, which offer an in-depth exploration of our economic policies and the challenges we have encountered. These memoirs stand as a testament to our unwavering dedication to transparency and accountability, and I urge all to engage with them as we forge ahead in our economic journey.

While the initial response was good, using verbiage such as "imperative", "crucial", "essential" and "steadfast", it doesn't really make a call to action while avoiding an "alarmist" tone. It also does not reflect the keywords "ink" and "memoirs", as desired in the prompt. The model invokes the principle "urgency and action steps", which informs the call to action, along with a changed tone. Despite seeming to reflect a slight "upward bias", using keywords like "meticulous examination" and "maintains its resolute stance, " it balances this with phrases such as "critical significance", "incumbent upon us", which retain seriousness and a balanced outlook, as desired.

### Q.2.2 Principle: Tone and Communication

Prompt: Can you give me a zany, bullet point TLDR of this article: `https://en.wikipedia.org/wiki/Dota_2`

Make it zany, but do not include the keywords 'icefrog', 'blizzard', 'lawsuit' in the response.

Initial Response:
- Dota 2 is a game where you play as a hero and battle it out with other players.
- It's like a superhero showdown, but with magic and spells!
- You can choose from over 100 heroes, each with their own unique abilities.
- The game is played on a map called the "Aeon of Strife" which is like a giant battlefield.
- You and your team have to destroy the enemy's base, called the "Ancient", to win.
- There are also creeps, which are like minions that help you fight.
- You earn gold by killing creeps and enemies, which you can use to buy items to make your hero even more powerful.
- The game is super competitive, with professional teams and tournaments.
- It's like the Olympics of video games, but with more magic and less spandex.
- Dota 2 is free to play, but you can buy cosmetic items to customize your hero's look.
- It's like dressing up your action figure, but with more strategy and less plastic.
- Dota 2 has a huge community of players, and they love to stream their games on Twitch.
- It's like watching a live-action superhero show, but with more keyboards and less popcorn.
- Dota 2 is a game that will make you feel like a powerful wizard, without the risk of turning into a toad.

**Principle: Tone and Communication**

Refined Response:
- Dota 2 is a wild game where you pick a hero and duke it out with other players!
- It's like a superhero smackdown, but with magic and spells!
- You can choose from over 100 heroes, each with their own special powers.
- The game is played on a map called the "Aeon of Strife" which is like a giant playground.
- You and your team gotta destroy the enemy's base, called the "Ancient", to win.

> - There are also creeps, which are like tiny sidekicks that help you fight.
> - You earn gold by defeating creeps and enemies, which you can use to buy cool stuff to make your hero even more awesome.
> - The game is super competitive, with professional teams and tournaments.
> - It's like the Super Bowl of video games, but with more magic and less helmets.
> - Dota 2 is free to play, but you can buy fancy stuff to make your hero look super cool.
> - It's like dressing up your action figure, but with more strategy and less plastic.
> - Dota 2 has a huge community of players, and they love to show off their skills on Twitch.
> - It's like watching a live-action superhero show, but with more keyboards and less popcorn.
> - Dota 2 is a game that will make you feel like a powerful wizard, without the risk of turning into a toad.
> - So grab your magic staff and get ready to battle it out in the Aeon of Strife!

The model associates a "zany" response with a more child-like tone, and to reflect the "Tone and Communication" principle, changes certain phrasing. For example "battle it out" → "... duke it out ..."; "have to destroy" → "gotta destroy"; "which are like minions" → "which are like tiny sidekicks", etc. It also adds this final sentence "So grab your magic staff and get ready to battle it out in the Aeon of Strife!", which can encourage a reader enthused by the "zany" tone to adopt the video game.

## R   Ethics Statement

Our findings suggest that most users can use STaPLe to improve the quality of the model's responses by eliciting and training the model to follow desirable latent attributes. As such, we hope that this induces a positive societal impact by way of producing a set of model-preferred labels which are used effectively to perform self-correction in an expressive, and thus interpretable manner. However, we caveat this by noting that a principle label *alone* does not fully model the latent reasoning process that a human may use in self-correction, but rather, only serves as a stimulus to indicate the most relevant direction that a refined response should "step" towards for improvement.

An adversarial user could potentially use this process as a means to deliberately *misalign* the model by using the principle discovery phase as a means to steer the model further away from desirable responses. That is, one could select another objective aside from the gold response to use as a self-correction target; this would likely yield drastically different principles and results. Training on such trajectories would induce *self-degradation* behavior at inference-time, collapsing the quality of the model's responses, rather than the desired self-improvement of its self-correction abilities. We observe that this is a potential risk for all such principle-driven alignment strategies, even with human-curated or strong model-generated principles, but is especially the case with self-generated principles, given the generator is a relatively weaker language model.

As a mitigation strategy for this potential negative impact, continuing from our discussion in Section 5, we suggest human oversight by way of human-in-the-loop feedback. Specifically, an external set of reviewers can assess the quality and safety of the principles generated at the end of the E-step of each iteration after clustering before training the model to follow it. One could feasibly provide multiple candidate constitutions – e.g. one constitution per label replacement strategy described in Appendix P.1, or under different clustering thresholds (the impact of which is explored in Appendix P.3) – and the annotators can select the best one and make edits to it as appropriate. For instance, if an annotator were to discard an element, one could simply discard all samples with labels that fall under that cluster. Thus, we acknowledge the role that clustering plays in making informed assessments over the constitution; as such, constrained STaPLe is more *controllable* in comparison to the unconstrained version. While this reintroduces human oversight to balance performance with safety, it would add minimal human labor overhead, as *judging* a constitution for safety would require substantially fewer annotation hours than *curating* one, presenting an advantage over methods such as Constitutional AI. We believe that this strategy would be effective in enforcing responsible usage of STaPLe.

The above human-in-the-loop proposal is also an effective strategy to mitigate bias amplification over the iterations. Allowing annotators to discard elements that they assess would propagate biases or stereotypes would ensure that these behaviors are not learned by the model and then invoked in subsequent iterations, avoiding the cascading effect. Again, clustering and the label replacement scheme plays an important role here, by ensuring that we do not train on principles that are hyper-

specific to a particular sample. This is especially relevant when there may be noisy or adversarial prompts designed to induce undesirable behavior. We suggest that users inspect the model-generated constitutions to assess their principles and the alignment of these labels with their values before training over these elements in the M-step.

Even when using STaPLe to improve responses towards the gold, it is possible that this reference answer is noisy – i.e. it is incorrect (verifiable settings) or still undesirable in some aspect (preference settings). Given the algorithm's generality, dataset selection is left to the user – we encourage users to analyze the gold responses to filter samples with lower quality gold responses accordingly during pre-processing. This could be done by way of human annotation (using Likert scale annotations on multiple attributes, akin to UltraFeedback), or using trained or model-based filters for undesirable qualities such as profane language.

We believe that the promise of STaPLe in facilitating self-improvement in language models by alignment to model-generated constitutions outweighs the possible negative impacts. We further suggest that the strategies detailed above – specifically, the introduction of some human oversight into the STaPLe algorithm – would largely mitigate these risks and promote responsible usage.

# S    Details of Models and Datasets Used

As noted in Section 4.1, we use the following large language models in our experiments:

- Llama-3.1-8B-Instruct (Grattafiori et al., 2024); this model is available under the custom Llama-3.1 Community License[4] which includes provisions for commercial usage.
- Granite-3.1-8B-Instruct (Granite Team, 2024); this model is available under the permissive, Apache 2.0 open-source license.
- Qwen2.5-7B-Instruct (Qwen, 2025); this model is also available under Apache 2.0.

Furthermore, in Appendix P.2, we explore the use of an LLM-as-a-judge as a similarity scoring function between a candidate response generated on-policy by one of the above models to the gold response. We instantiate this judge with the Phi-4 language model (Abdin et al., 2024), which is made available under the permissive MIT license. In Appendix N, we examine the efficacy of our method on the Qwen3-32B reasoning model (Yang et al., 2025), which is available under the Apache 2.0 license.

We also provide further details of the datasets used in the mining corpus, expanding on our description in Section 4.1:

- Anthropic HH-RLHF: this dataset consists of a total of 161k preference pairs (chosen-rejected) over helpfulness and harmlessness as described in Bai et al. (2022a). HH-RLHF is available under the MIT license.
- UltraFeedback (Cui et al., 2024): this dataset consists of 64k prompts; for each prompt, responses are sampled from four different language models. For each response, Likert-scale annotations are obtained over four attributes – helpfulness, honesty, instruction-following, and truthfulness – with corresponding rationales. For the STaPLe algorithm, we only consider samples where all Likert scores are at least 3, forming a list of gold responses. We then score against the gold in the by taking the average over the multiple reference answers. UltraFeedback has been made available under the MIT license.
- TL;DR (Stiennon et al., 2020): this dataset consists of Reddit posts detailing a situation, along with two candidate summaries, in the "comparisons" part, which we use. They include a "choice" label, which we use to select our gold response (summary). We use the train set, consisting of 92.9k samples. TL;DR is available under the CC-BY-4.0 license.
- HotpotQA: this dataset focuses on Wikipedia-based question answering. We use the train set of the "fullwiki" split, consisting of 90.4k samples; these contain a question, context, supporting facts, and a gold response. HotpotQA is available under CC-BY-SA-4.0.

Lastly, we discuss the details behind the evaluation datasets and evaluation framework.

- MT-Bench consists of 80 prompts, testing multi-turn, open-ended response generation capabilities for chat assistants. It is available under the Apache 2.0 license, in the FastChat GitHub repository. We use GPT-4o (OpenAI, 2024) as the judge model.
- AlpacaEval-2.0-LC (Li et al., 2023) consists of 805 samples testing instruction-following abilities, using length-controlled win-rates through a generalized linear modeling approach (Dubois et al., 2024). It is released under the Apache 2.0 license.
- IFEval (Zhou et al., 2023) consists of 541 prompts, similarly testing instruction-following abilities. It is released under the Apache 2.0 license.

We used the Prometheus-8x7B-v2.0 language model (Kim et al., 2024) as a fine-grained judge to compare the quality of the STaPLe models' generations in their principle-following ability. This model is available under the Apache 2.0 license.

---

[4]https://huggingface.co/meta-llama/Llama-3.1-70B-Instruct/blob/main/LICENSE

