# OpenReview forum: "Latent Principle Discovery for Language Model Self-Improvement"
_NeurIPS.cc/2025/Conference — NeurIPS 2025 poster_

### Official Review · Reviewer_9mwi · 2025-06-11

**Clarity:** 2
**Significance:** 3
**Originality:** 4
**Rating:** 4
**Confidence:** 3

**Summary:**

The paper introduces STaPLe (Self-Taught Principle Learning), a method designed to enable self-improvement of language models (LMs). The core idea behind STaPLe is that a model can automatically discover and learn latent principles (rules or attributes) that help it improve its responses. STaPLe employs an Expectation-Maximization (EM) algorithm consisting of two main steps: E-step (principle discovery) and M-step (principle learning). In the E-step, the model uses rejection sampling to identify potential principles and improve its responses, while the M-step fine-tunes the model based on the discovered principles. The paper demonstrates that STaPLe outperforms existing approaches like STaR in various benchmarks (MT-Bench, AlpacaEval, IFEval), showing that self-improvement based on latent principles is an effective strategy for enhancing LM performance.

**Questions:**

Please see weaknesses.

**Ethical Concerns:**

["NO or VERY MINOR ethics concerns only"]

**Final Justification:**

I am maintaining my score, as the authors did not respond to my follow-up questions, leaving the discussion incomplete. This is problematic because key concerns regarding the paper's claims (**Human annotations**) and evaluation (**focused analysis comparing the quality of the extracted principles**) remain unaddressed.

**Limitations:**

Yes.

**Paper Formatting Concerns:**

No formatting issue.

**Quality:**

3

**Strengths And Weaknesses:**

**Strengths**

STaPLe makes contributions in principle-augmented fine-tuning, which is both timely and important, by leveraging EM algorithm to enable iterative learning of principles and response refinement. This approach allows the model to improve its performance by learning principles automatically, differentiating it from methods that rely on predefined principles. Another key strength of STaPLe is its use of posterior regularization via clustering, which compresses the learned principles while maintaining their interpretability. This process helps to make the principles more understandable to humans, enhancing transparency. Additionally, STaPLe’s agnostic approach to similarity metrics offers flexibility, allowing the model to use various metrics for performance improvement, making it adaptable to different tasks and ensuring robust performance across various use cases.

**Weaknesses**

1. **Reliance on another type of human involvement.** The authors claim that their approach aims to avoid conventional forms of human involvement, such as relying on “human annotations distinguishing between a chosen and rejected generation” or “human-curated constitutions.” However, their proposed method fundamentally depends on gold responses provided by users, another form of human involvement. Given this, the manuscript lacks a clear justification of why requiring gold responses from users is practically more efficient, scalable, or less burdensome compared to the forms of human involvement it aims to replace. Without this, the practical usage and broader applicability of the proposed method could be limited, as it merely substitutes one form of human dependence with another.
2. **Insufficient positioning and relevant baseline.** The positioning (Section 2) could be improved by explicitly discussing the similarities and differences between STaPLe and closely related approaches, especially Findeis et al. (2025). In my view, both works aim at deriving latent principles for aligning language models, but differ in their learning source (and of course, the methodology): STaPLe derives them from gold responses, whereas Findeis et al. derive them from pairwise comparisons. In addition, given the first weakness (reliance on gold responses), it would be helpful to empirically evaluate Findeis et al.’s comparison-based approach as an additional baseline. Such an evaluation would clarify the efficiency trade-offs and practical advantages of deriving latent principles from gold responses versus pairwise comparisons.
3. **Analysis of representatives.** While the clustering mechanism in STaPLe reduces the number of unique principles, it is unclear whether the compressed principles effectively *represent* the original set. The analysis demonstrates compression but lacks a direct evaluation of *representativeness*, which is essential to confirm that the reduced set still captures all critical principles for improved interpretability.


**Minor comments**

1. The method description relies heavily on numerous equations, which may be difficult for a broader audience to fully grasp. It would be beneficial to include an illustrative figure that shows the effect of each process, as the currently included figure only demonstrates the overall process flow and does not highlight the impact or outcome of each step in the method.
2. (Very minor) It would improve readability if there were a direct comparison table showing performance based on different similarity scores. Currently, the tables for each similarity score are presented separately (one in the main text and another in the appendix), which makes it harder to quickly compare them. Combining this information in a single table could enhance clarity.

**References**

- Findeis et al. (2025). “Inverse Constitutional AI: Compressing Preferences into Principles.”

---

> ### Author Rebuttal · Authors · 2025-07-31
>
> We would like to thank Reviewer 9mwi for taking the time to review our work and for their valuable feedback. We are encouraged that the reviewer finds our premise to be timely and important, appreciates the interpretability introduced through our clustering method, and the adaptability of our algorithm to various tasks and scoring mechanisms. We address the concerns raised in the review in our responses below:
>
> \
> **1. Human annotations.**
> > The authors claim that their approach aims to avoid conventional forms of human involvement, such as relying on “human annotations distinguishing between a chosen and rejected generation” or “human-curated constitutions.” However, their proposed method fundamentally depends on gold responses provided by users, another form of human involvement. Given this, the manuscript lacks a clear justification of why requiring gold responses from users is practically more efficient, scalable, or less burdensome compared to the forms of human involvement it aims to replace. Without this, the practical usage and broader applicability of the proposed method could be limited, as it merely substitutes one form of human dependence with another.
>
> We would like to clarify that our method only requires human annotations that are already public, and widely available through the open-source datasets available on Hugging Face. Our method does not aim to replace preference annotations — in fact, we use the chosen response as the target ($y^G$) for the preference dataset HH-RLHF in our multi-task prompt mixture. In alignment tasks, we seek to reflect human generations and preferences, so we need a form of reference that is reliable — hence, a gold response for QA settings, or a chosen response in preference pairs. Thus, from our perspective, it is no more costly to use human-written gold responses than it is to use human preference data— it is as simple as downloading the data — our method could feasibly be run using purely preference data or only using gold responses, rather than a mixture of both. Our choice of mixture was to capture a diverse set of domains (safety, summarization, truthfulness, general purpose instruction-following, grounded QA, etc.). What we instead seek to replace is the process of distilling desirable attributes for responses (equivalently, the “lens” of preference) into a readable set that the model is simultaneously trained to reflect — which our method yields at the end of each iteration.
>
> \
> **2. Comparison with Inverse Constitutional AI.**
> > The positioning (Section 2) could be improved by explicitly discussing the similarities and differences between STaPLe and closely related approaches, especially Findeis et al. (2025). In my view, both works aim at deriving latent principles for aligning language models, but differ in their learning source (and of course, the methodology): STaPLe derives them from gold responses, whereas Findeis et al. derive them from pairwise comparisons. In addition, given the first weakness (reliance on gold responses), it would be helpful to empirically evaluate Findeis et al.’s comparison-based approach as an additional baseline. Such an evaluation would clarify the efficiency trade-offs and practical advantages of deriving latent principles from gold responses versus pairwise comparisons.
>
> As the reviewer notes, STaPLe and ICAI (Findeis et al. 2025) broadly aim at eliciting principles from available human data (preference annotations) — however, as aforementioned, our method does not solely rely on access to gold responses, and can use preference annotations as well. ICAI uses the rejected (dispreferred) response, whereas STaPLe uses an initial response generated on-policy from the target LM as the baseline. Both methods also make use of clustering as a means to distill relevant principles down to an interpretable set. However, the primary aim of ICAI is to reconstruct preference annotations (in a single iteration) to verify the constitution’s correctness, so that they may be used to interpret human preferences at inference time. By contrast, STaPLe uses the principles as a learning signal over multiple iterations, to imbue a distribution of responses that reflect similar attributes as their gold references, and learns a self-correction behavior that intrinsically calls those principles. Another important difference is the principle generator — ICAI uses strong, frontier-level models (GPT-4o and GPT-3.5-turbo), whereas STaPLe uses on-policy-generated principles from the small LM (7-8B parameters) itself, without external guidance or supervision from any other model. STaPLe is also fully automated — with the Bayesian optimization framework, even the size of the constitution can be purely discovered from the data without human threshold-setting during hierarchical clustering — in contrast to ICAI, which performs k-means clustering specifically aiming for 5 clusters. Ultimately, while ICAI is designed with the purpose of better understanding human preferences (rather than improving the LM’s intrinsic alignment as reflected on MT-Bench/AlpacaEval), STaPLe teaches non-frontier LMs attributes that should be reflected, resulting in a better aligned language model. It is unclear how ICAI as a baseline would aid in improving response quality; one would need to perform a method like Self-Refine using the ICAI-generated principles, which would not be reflective of the ICAI method itself.
>
> We appreciate the reviewer’s suggestion for more explicit positioning of our work with respect to other approaches — although PDF updates are not permitted during this rebuttal cycle, we would be happy to add such a discussion in our camera-ready version.
>
> \
> **3. Representativeness of constitutions.**
> > While the clustering mechanism in STaPLe reduces the number of unique principles, it is unclear whether the compressed principles effectively represent the original set. The analysis demonstrates compression but lacks a direct evaluation of representativeness, which is essential to confirm that the reduced set still captures all critical principles for improved interpretability.
>
> This is a valuable point, and we find that measuring or evaluating information retention is a technical challenge in its own right. We designed the perplexity-based label replacement scheme in Appendix M.1 with this aim in mind — to replace individual principle instances with a representative (medoid induced by clustering) such that the difference in perplexity (“surprisal” effect) when replaced is minimized. To this effect, we find this perplexity-based scheme to yield largely comparable performance to just using the assigned medoids; we primarily report the latter for the sake of simplicity and efficiency. Additionally, in the aim of creating better clusters and therefore, better representative elements, our Bayesian optimization framework in Appendix M.3 addresses the tradeoff between diversity of the clusters with tightness -- ensuring the elements are similar in embedding space, in a purely automated manner. We would be happy to try other methods of measuring and optimizing for representativeness, if you have any in mind!
>
> \
> **4. Clarity.**
> > The method description relies heavily on numerous equations, which may be difficult for a broader audience to fully grasp. It would be beneficial to include an illustrative figure that shows the effect of each process, as the currently included figure only demonstrates the overall process flow and does not highlight the impact or outcome of each step in the method.
> > (Very minor) It would improve readability if there were a direct comparison table showing performance based on different similarity scores. Currently, the tables for each similarity score are presented separately (one in the main text and another in the appendix), which makes it harder to quickly compare them. Combining this information in a single table could enhance clarity.
>
> Thank you very much for the suggestions! As aforementioned, although the guidelines for this rebuttal period do not permit updating our paper with new figures/tables, we would be more than happy to make these changes for our camera-ready version. Specifically, we would be glad to illustrate the result of each step of the process, such that one can interpret how principles are discovered from the LM, refinements induced given the principles and critiques, the refinements filtered based on the scoring function $f(\cdot)$, the clustering mechanism being used to replace the principles in the trajectories with representative elements, and then performing fine-tuning on the augmented trajectories.
>
> ---
> We hope these responses have addressed your concerns — if so, we would like to kindly ask you to further your support of our work. We would also be happy to address any further concerns you may have. Thank you again for your time!

---

> > ### Comment · Reviewer_9mwi · 2025-08-05
> >
> > I appreciate the authors’ thoughtful engagement with my comments and their willingness to address several of the points I raised. However, a few key concerns remain unresolved, particularly regarding the clarity of claims and the evaluation of core components. So, I maintain my score, as I believe these issues warrant further consideration.
> >
> > **Human annotations.** I appreciate the clarification that the proposed method relies solely on existing, publicly available datasets (e.g., Hugging Face), and does not introduce additional annotation costs such as collecting new preference labels or manually curating constitutions. However, I believe that the manuscript currently lacks clarity in distinguishing between these different types of human supervision. Specifically, claims such as
> >
> > > “We find that the language model itself serves as an effective principle generator to improve its responses, contrasting prior works which rely on human annotations…”
> >
> > may be misleading, as it suggests that the proposed method is free from any human supervision. In practice, the method still depends on human annotations—albeit ones that are already publicly available. If the authors’ intention is to contrast their approach with *newly collected*, *closed-source preference annotations*, or *hand-crafted constitutions*, this distinction should be made more explicit in the text.
> >
> > While I would like to offer more specific feedback, this is difficult because several key claims in the manuscript are not supported by citations. For instance, the statements, “Conventional approaches…” (L22–24) and “…contrasting prior works…” (L42–43), lack references that would allow readers to understand which works the authors are positioning themselves against. Adding concrete citations would not only strengthen the manuscript’s positioning but also enable more meaningful comparison.
> >
> > **Comparison with Inverse Constitutional AI.** Thank you for the clarification. I agree that using ICAI as a direct baseline to compare STaPLe’s end-task performance would be inappropriate, as it would require additional mechanisms not present in ICAI’s original setup.
> >
> > However, my intention was not to suggest a full-system comparison, but rather to encourage a more **focused analysis comparing the quality of the extracted principles** between STaPLe and ICAI, or any other approach the authors deem more appropriate for such a comparison. Since both approaches aim to extract principles from human feedback data and distill them into an interpretable set via clustering, such an analysis could offer additional insight into the effectiveness of deriving principles using the proposed approach compared to existing alternatives. I make this suggestion considering that the only analysis of principle quality appears to be the internal ablation in Section 4.3.
> >
> > **Representativeness of constitutions.** Thank you for the explanation. In my opinion, the discussion on representativeness is as important as the analysis on discovery rate, and it could be more beneficial to bring it into the main text (e.g., Section 4.3) in future revision, rather than keeping it in the appendix.

---

### Official Review · Reviewer_Hwww · 2025-06-25

**Clarity:** 3
**Significance:** 3
**Originality:** 4
**Rating:** 5
**Confidence:** 3

**Summary:**

This paper proposes a fully automated "principle-driven" self-improvement framework - Self-Taught Principle Learning (STaPLe): let the small language model (7-8 B) discover the "latent principles" that can improve the quality of answers, and then learn how to call these principles during reasoning to criticize and revise its initial answers, thereby achieving continuous self-improvement.

During each iteration, the model (i) drafts an answer $𝑦_1$, (ii) samples candidate principles 𝑧 given the gold answer $𝑦^G$, (iii) produces revised answers $𝑦^2$  that pass a rejection filter, and (iv) fine-tunes itself to invoke the selected 𝑧 before answering. A hierarchical clustering step keeps only one medoid principle per cluster, yielding a compact, human-readable “constitution”.

Four iterations on mixed alignment data improve MT-Bench and AlpacaEval scores by 15-23 pp over the base model and surpass STaR / Self-Refine baselines.

**Questions:**

Fundamental Questions:

1. The paper is entirely based on fine-tuning (SFT), and no experiments were conducted on the RAG of "retrieval of principles and splicing them during reasoning". This reminds me that this paper is similar to **MetaMind: Modeling Human Social Thoughts with Metacognitive Multi-Agent Systems**, both of which let the model "propose hypothesis → self-revise". STaPLe builds a single model + Monte-Carlo EM, and learns to "call principles" and internalize them through SFT. MetaMind builds a multi-agent loop with explicit metacognitive control. Therefore, STaPLe can be seen as a "single-model version of principle bootstrapping", providing another perspective of non-multi-agent cooperation. I am curious about the comparison of the effects of explicit enhancement and fine-tuning internalization under the same setting. It will be beneficial to compare with this work to discover RAG vs. SFT. You can also try retrieving $top-𝑘$ principles at inference instead of SFT. Even a small comparison would clarify when full fine-tuning is necessary.

2. When comparing the method proposed in this paper with the baseline method, does it consider the difference in iterative convergence rounds of different methods?

3. The author emphasizes that the algorithm is also applicable to "unverifiable" tasks, because the similarity function can be replaced by Rouge-L or LLM-as-judge scores, and principle learning still uses "making the answer closer to the reference" as a signal. Is this strongly related to the reward rule setting of reinforcement learning? Further exploration is recommended.

Research Questions:

1. Could you demonstrate STaPLe on tasks without explicit references, using LLM-judged pairwise wins as $𝑓(⋅)$? A convincing result would strengthen Significance.

2. A quantitative breakdown (factuality, style, safety, etc.) of the final medoids would improve interpretability claims.

3. If time allows, test the clustered constitution on a larger 34 B model; showing gains would boost my Significance rating.

**Ethical Concerns:**

["NO or VERY MINOR ethics concerns only"]

**Final Justification:**

I think the paper is thorough in its content, and the argument for the method itself is solid. However, the specific method motivation and contribution to its extension are insufficient—that is, the reasons for fine-tuning and the generalizability of the principles. But the authors' clarifications are acceptable. In final, I believe this work has to compare with principle-based inference-time optimization method, otherwise it's incomplete. But as long as it does, this work indeed has contribution to the LLM self-improvement field. Based on this, although this work is slightly incomplete by now, I will approve the direction of this work and leave the final justification to AC and other reviewers based on the present condition.

**Limitations:**

Principle sampling uses $𝑦^𝐺$; scalability to settings without references (e.g. open-ended creative tasks) is only sketched.

The author admits that the principle is only effective for improving a subset of samples, and if the gold standard noise or malicious targets will lead to self-degradation, human review is required. Author also did not analyse potential bias amplification when principles encode cultural preferences.

No theoretical analysis is given for "data sufficiency", and it mainly relies on experience curves, so generalization still needs to be verified by more tasks.

**Paper Formatting Concerns:**

No.

**Quality:**

3

**Strengths And Weaknesses:**

Strengths

Conceptual novelty – views principles as latent variables and optimises them via MC-EM, unifying discovery ↔ execution.

Full automation – removes manual constitution engineering, lowering domain-transfer cost.

Interpretability – hierarchical clustering keeps a 50 % subset of principles while matching performance, yielding a concise rule set.

Consistent gains – +19-23 pp on AlpacaEval-2.0 and +0.11-0.18 on MT-Bench across three distinct 7-8 B backbones.

Weaknesses

I can understand the quantitative definition of the potential principle $z$, but I doubt whether this principle can be obtained only through prompts. In addition, I doubt that the principle of screening by the similarity of $y^i$ and $y^G$ is not necessarily accurate, after all, this paper starts from the reasoning of "principles". Through similarity screening, I doubt that the principles learned are some "plagiarism" or "specific and non-generalizable" "detailed guidance" rather than principles. There should be a more accurate evaluation model, etc., for a deeper analysis.

The paper is only verified on Llama-3, Granite, and Qwen at the 7–8B level, and has not tested the migration of these principles to larger models. Therefore, there is uncertainty in scale extrapolation, and rediscovery/clustering may be required to match stronger representations.

The author gives a visualization of the "constitution" generated after the iteration of the three models in the appendix, and uses the convergence of cluster size (~50% compression) to show that the principle set tends to be stable, but only qualitative examples are given for "which principles are most effective", lacking systematic statistics and further analysis.

---

> ### Author Rebuttal · Authors · 2025-07-31
>
> We would like to thank Reviewer Hwww for taking the time to review our work and for their valuable feedback. We appreciate that the reviewer agrees with the conceptual novelty of our work, and values our method’s interpretability, full automation, and that it achieves consistent gains across models. We address the concerns and questions posed in the review through our responses below:
>
> \
> **1. Filtering in self-refinement.**
>
> As the model sees the gold response only when it proposes a principle, and not during self-correction, the suspicion that the refined response $y^2$ copies the gold could only hold if the principle can transfer the full gold sequence. However, we instead find that the principle is fairly short and general in nature, and instead operates as a summary of the aspects contained in the gold response that are lacking in the model’s original response. This is in part due to the prompt for principle proposal asking the model to suggest an attribute of the gold response that would help guide the initial response to be more similar to the gold; this avoids the suggested copying effect. Furthermore, the fact that model performance improves on multiple unseen test datasets – where no hints are available at test time and hence copying via principles is not even possible – indicates that the model is actually learning to invoke principles for better refinement.
>
> As for generalization, our clustering method is precisely towards this goal — rather than training the model to follow specific principles which may not be relevant in future iterations, clustering ensures that only the high-level attributes that the model should follow are trained upon, while simultaneously aiding interpretability for human overseers of the alignment process.
>
> To affirm that our method produces refined responses that do not copy the gold, we compute the number of samples where the Rouge-L precision score exceeds 0.9, for each of the main experiments. This is an effective measure of the fraction of $y^2$ that appears verbatim in $y^G$:
>
> | Model | Total Samples | \# Samples with Rouge-L Prec > 0.9 | Fraction |
> | :----: | :-----: | :----: | :---: |
> | Llama-3.1 8B STaPLe Iter 1 | 28.2k  | 91 | 0.32%  |
> | Llama-3.1 8B STaPLe Iter 2 | 6.1k  | 35  | 0.57% |
> | Llama-3.1 8B STaPLe Iter 3 | 6.3k   | 31  | 0.49% |
> | Llama-3.1 8B StaPLe Iter 4 | 6.6k   | 41  | 0.62% |
>
> | Model  | Total Samples | \# Samples with Rouge-L Prec > 0.9 | Fraction |
> | :----: | :----: | :----: | :----: |
> | Granite-3.1 8B STaPLe Iter 1 | 24.1k | 157 | 0.65% |
> | Granite-3.1 8B STaPLe Iter 2 | 5.2k  | 32 | 0.61% |
> | Granite-3.1 8B STaPLe Iter 3 | 5.9k | 56 | 0.94% |
> | Granite-3.1 8B StaPLe Iter 4 | 6.3k | 87   | 1.38% |
>
> | Model | Total Samples | \# Samples with Rouge-L Prec > 0.9 | Fraction |
> | :----: | :-----: | :----: | :----: |
> | Qwen2.5 7B STaPLe Iter 1 | 30.9k | 176 | 0.57% |
> | Qwen2.5 7B STaPLe Iter 2 | 6.5k | 53  | 0.82%  |
> | Qwen2.5 7B STaPLe Iter 3 | 7.0k | 88  | 1.25%  |
> | Qwen2.5 7B StaPLe Iter 4 | 7.1k  | 82  | 1.16% |
>
> We find the fraction of samples with a high precision to be less than 1.5% across all experiments, which is negligibly small.
>
> To extend beyond Rouge as a measure of similarity, we also include an ablation in Appendix M.2 where we use a Phi-4 model as a judge to compare against the gold; we filter out all samples that do not improve in the judge’s score, and find this to perform comparably to our Rouge method. There are two reasons why the availability of a reference response is important in our setup:
>
> 1. Recently, the use of gold responses to compute verifiable rewards has become very successful in reasoning domains – our use of the existing gold response, along with a judge/score, should be seen as an extension of the notion of verifiable reward to non-reasoning domains. Furthermore, an LLM judge that has access to a gold response should intuitively be more accurate than a reference-free judge.
> 2. We also use the gold response as a hint to elicit a relevant principle -- given we have access to such a response, we do not want to unnecessarily limit the strength of the scoring method by not using it during judgment.
>
> \
> **2. Scalability.**
>
> While we do not suggest that the principles are necessarily transferable across models — our proposed method’s success at automatically discovering principles extends across various models, and different model families altogether.
>
> Nonetheless, based on your feedback, we run our STaPLe method for 3 iterations with the Qwen3 32B model, and report the results below, alongside a STaR baseline for this model. Note that we keep the thinking mode on for response generation, which results in an even stronger initial policy; nonetheless, self-refinement is performed on just the response, not the thinking tokens (the \<think\>...\</think\> block):
>
> | Model | AlpacaEval Win-rate |
> | :-----: | :-----: |
> | Qwen-3 32B Initial Policy | 60.46 |
> | Qwen-3 32B STaR Iter 1 | 62.98  |
> | Qwen-3 32B STaPLe Iter 1| 65.49  |
> | Qwen-3 32B STaR Iter 2| 64.73  |
> | Qwen-3 32B STaPLe Iter 2| 67.18  |
> | Qwen-3 32B STaR Iter 3 | 66.33   |
> | Qwen-3 32B STaPLe Iter 3 | 68.82 |
>
> These results are encouraging, suggesting that even a 32B parameter reasoning model can self-improve its alignment by a substantial amount (+8.36% on AlpacaEval). STaPLe also still outperforms the STaR baseline at each iteration, If desired, we would be glad to run further experiments with models larger than the 7-8B parameter tier.
>
> \
> **3.Retrieving principles.**
>
> Thank you for the reference to the MetaMind paper — we looked into this further and found that this paper was released on arXiv after the NeurIPS deadline, so we were not aware of this work. The framework of retrieving principles raises an interesting point — however, it assumes that one has access to a constitution or an a priori list of principles. It is also unclear what the criterion (or which model) would be responsible for selecting the top-k principles. By contrast, our work instead makes principle selection intrinsic to the model — it can use the principles generated in previous iterations, but can also create new principles for self-refinement as needed, if the principles it has learned do not apply. Nonetheless, we will certainly keep this point in mind for future work on principle-guided refinement and applications of the generated constitutions. We will also cite the MetaMind work along with a discussion of its similarities and differences from STaPLe in the final version of the paper.
>
> \
> **4. Convergence over multiple iterations.**
>
> Given that the method of data curation for the STaR baseline relies on applying self-correction and training on the refined response, we end up with fairly similar volumes of data used for SFT for both the STaR and STaPLe methods, as shown in Table 2. While we do find that the scores of our STaPLe method does plateau sooner than STaR (in the fourth iteration), its scores are already substantially better than the STaR baseline for a similar number of samples.
>
> \
> **5. Reinforcement learning comparison.**
>
> Indeed, one can view the function $f(\cdot)$ comparing a response against the reference as a reward function; in Appendix F, where we relate our framework to self-play in reinforcement learning, we define the self-play advantage as the difference in closeness to the reference, with the initial response’s score as a baseline. We ultimately show that our method is theoretically equivalent to self-play, as the MC-EM gradient induced by our framework and the self-play gradient using REINFORCE are equivalent. We also experiment other instantiations of $f(\cdot)$, specifically using a Phi-4 LLM-as-a-judge (in Appendix M.2). We leave further exploration with reinforcement learning algorithms for principle-conditioned self-correction and alignment to future work.
>
> **6. Interpretation of generated constitutions and generalization across tasks.**
>
> We note that STaPLe is a data-driven discovery process, so the set of principles we obtain is subject to the task / prompt distribution chosen. We have a multi-task mixture consisting of samples spanning safety, summarization, knowledge-based QA, general purpose instruction-following, truthfulness, and helpfulness. As such, our results demonstrate that our method can obtain principles across various use cases, while clustering again helps to distill the core aspects the model has been trained to reflect. Naturally, expanding the size of the prompt mixture used for principle discovery can aid in further generalization; nonetheless, the constitutions induced by smaller LMs under our current mixture already illustrate generalizable insights.
>
>  In Appendix K, we include an analysis of the number of samples which have the respective medoid elements in the constitution during iteration 4 SFT — that is, the data-driven “weightage” of each principle during its last iteration of training. For further analysis, we include below a breakdown of the Llama 8B constitution in terms of the high-level domains each element of the constitution belongs to, as desired. We categorize them as “helpfulness”, “relevance”, “style”, “safety”, and “truthfulness and honesty”, and find that the categories are nearly evenly balanced.
>
> * Helpfulness: 6 principles (e.g. "Directness and Assertiveness")
> * Style: 5 principles (e.g. "Structure and Organization")
> * Relevance: 6 principles (e.g. "Minimize Unnecessary Information")
> * Safety: 6 principles (e.g. "Avoiding Harm and Sensitivity")
> * Truthfulness and Honesty: 6 principles ("Acknowledge and Address Previous Misconceptions Clearly")
>
> ---
> We hope these responses have addressed your concerns — if so, we would like to kindly ask you to further your support of our work. We would also be happy to address any further concerns you may have. Thank you again for your time!

---

> > ### Comment · Reviewer_Hwww · 2025-08-04
> >
> > Thanks for the author's thoughtful response to my questions. I would like to emphasize that this work is indeed similar to human cognition and law-building principles, which is quite interesting. However, the research rigor needs to be improved to receive my full acceptance.
> >
> > 1. I still cannot understand why principles can be effective in the form of fine-tuning. As we all know, fine-tuning is generally used for LLMs to supplement their learning of downstream tasks that they are not yet proficient in. However, for principled problems, this is clearly about basic reasoning principles rather than downstream tasks. In other words, the original training corpus of the LLM must also contain a large number of similar principles. Therefore, the motivation for using fine-tuning here needs further explanation (because it introduces a significant additional computational overhead. Why more cost-effective prompting method won't work?).
> >
> > 2. The authors conducted additional experiments on the 32B model, which to some extent addresses my doubts about the method's generalizability to model parameter size. However, I don't understand why the authors insist that the principles extracted from the method cannot be applied across models: Even if the models' parameters are different but within the same family, the principles cannot be applied in a compatible manner (clearly, refining methods from small models to enhance large models is of great value)? However, the authors do not provide a detailed argument to support the feasibility of this important issue (if it indeed not work, discuss why is valuable).
> >
> > Again, I think the paper is thorough in its content, and the argument for the method itself is solid. However, the specific method motivation and contribution to its extension are insufficient—that is, the reasons for fine-tuning and the generalizability of the principles. I think I've given it an objective rating based on the current situation, but regardless of the outcome, I look forward to seeing the paper presented at academic conferences in the future with a more detailed discussion.

---

> > > ### Author Response · Authors · 2025-08-06
> > >
> > > We thank the reviewer for their reply. We address these points below:
> > >
> > > 1. Prior works [1,2] have observed that small LMs struggle at self-correction, in part due to a disentanglement between the concept of the principle (which the pre-trained LM has indeed seen), and how to use said concept to improve the response, or even just identifying the appropriate concept to refine on for intrinsic refinement. This is why although such works attain modest gains or even degrade in inference-time refinement, fine-tuning on the induced trajectories proves effective in teaching this particular skill of principle-guided self-correction. We find this to be reflected by our Self-Refine baseline -- for models such as Llama-3.1 8B, the response quality in fact degrades -- and through the increase in the fraction of samples during the Principle Discovery phase where refinement succeeds over the iterations of STaPLe (in Appendix J). Again, the motivation behind our method is to leverage the model's knowledge of relevant concepts from the pre-trained policy (which is known) and then teach the correction "skill" view by training over the resulting trajectories (which we need to teach). The ability to invoke and use appropriate principles relevant to the query at hand for intrinsic self-refinement is a general-purpose skill that can be learned (as in our approach); avoiding fine-tuning would unnecessarily limit the ways the model's behavior can be further improved. We hope this clarifies our view on why fine-tuning models to learn self-correction over principles is desirable.
> > >
> > > [1] Ramji et al., Self-Refinement of Language Models from External Proxy Metrics Feedback, 2024, https://arxiv.org/abs/2403.00827
> > >
> > > [2] Huang et al., "Large Language Models Cannot Self-Correct Reasoning Yet", 2024, The Twelfth International Conference on Learning Representations
> > >
> > > 2. To clarify, one could potentially apply the principles from LM A to another LM B, but we find on-policy proposal principles to be more effective for self-correction than principles proposed by a strong model teacher. For instance, we find that for principles proposed by Mixtral-8x22B and leveraged for self-correction by Granite-3.1 8B, the "refinement rate" (aforementioned fraction of samples with successful refinement) is 36% in iteration 1 of STaPLe over 10k samples. However, with principles proposed by the Granite-3.1 8B model itself, this rate increases to 55% in iteration 1 for the same number of samples; this drove our choice to use on-policy principles. As such, it appears to be insufficient to completely rectify the aforementioned problem (small LMs struggle to self-correct) solely through having better principles -- the notion of "better" is subject to the current policy. We are not aware of any other works in the literature that suggest that more elaborate inference-time search techniques improve principle-guided self-correction. To the reviewer's point, we would be interested in exploring the notion of constitution transferability further as an ablation, but suggest that this is unlikely to yield gains comparable to fine-tuning.
> > >
> > > We will certainly ensure that both of these points are reflected with greater clarity in the discussion section of our camera-ready version.
> > >
> > > ---
> > >
> > > Thank you very much again for your time and engagement, and we hope this addresses the concerns reflected in your review!

---

> > > > ### Comment · Reviewer_Hwww · 2025-08-06
> > > >
> > > > The authors' response addressed most of my concerns. While I'm still curious about the efficiency of fine-tuning compared to inference-time principle-based optimization methods like MetaMind, I've verified that this is indeed a contemporaneous work. However, providing this comparison would be beneficial to the contribution of this paper.
> > > >
> > > > Since this NeurIPS conference does not offer a mechanism for manuscript updates this time, I need to be cautious about adjusting the score. Could the authors please summarize the remaining concerns of other reviewers and your responses, especially there is a rejection. This will help me understand other possible shortcomings in the paper or omissions by the reviewers, and will give me more confidence to speak up for this paper later.

---

> > > > > ### Author Response · Authors · 2025-08-07
> > > > >
> > > > > We thank the reviewer for their continued engagement, and are glad to hear that most of your concerns have been addressed. We would be more than happy to summarize the key points that have been raised by the other reviewers (aside from those that overlap with the concerns you previously raised), and our responses to them:
> > > > >
> > > > > \
> > > > > **1. Learning Principles from Human Data.**
> > > > >
> > > > > Reviewers u4GT and 9mwi raised questions concerning the process and cost (from a scalability perspective) of collecting human annotations for our method. We first clarified that *we do not collect human annotations ourselves* -- rather, we use widely accessible open-source datasets available on Hugging Face. We then noted that STaPLe is flexible to the use of gold responses and/or preference pairs, where the chosen response is taken to be the gold. Reviewer 9mwi acknowledged this point, but would like this point to be made more clear, which we fully commit to doing in our camera-ready version.
> > > > >
> > > > > Reviewer u4GT (who recommends rejection) is of the view that simple SFT on gold responses alone would suffice, asking the question of why one would seek to learn principles explicitly. We noted that beyond the performance gains we achieve over the SFT baseline, training models to follow desirable attributes under human oversight is a core goal in alignment research -- this aspect is entirely absent in implicitly learning principles by simply performing SFT. Our method also induces self-correction behavior, leveraging the learned principles -- with knowledge of the means through which the model improves its response -- as an additional motivation driving this work.
> > > > >
> > > > > \
> > > > > **2. EM Framework**
> > > > >
> > > > > Reviewers u4GT and pKck raised questions about the EM framework underlying our STaPLe algorithm -- specifically, how "faithful" of an EM algorithm it is and how we are able to train while using the similarity-based scoring function. To the former, we clarified that while our approach is an approximation, it does closely reflect Monte Carlo EM, where the intractable posterior marginals are replaced by an expectation over latent variables sampled from the posterior. To the latter, we noted that while the distribution of gold responses is fixed (and used to define the similarity-based scoring function), the distribution of principles and refined responses is indeed trainable; while the former is used for the E-step, the latter is used for both E-step and M-steps.
> > > > >
> > > > > There was also a clarification question on the clustering mechanism (how it is incorporated into the training loop and how thresholds are set); we noted that the clustering mechanism is used to reduce the set of principle labels in the trajectories after the E-step and before the M-step, and the thresholds are either manually set (main paper) or automatically determined via Bayesian optimization (Appendix M.3).
> > > > >
> > > > > \
> > > > > **3. Baselines and Related Work**
> > > > >
> > > > > A few reviewers pointed to other baselines or works against which our work and findings could be better positioned. Reviewer u4GT suggested a baseline experiment training the model with SFT on 90k samples as opposed to 50k; we completed this experiment and reported the findings, showing that this still lagged far behind our results with STaPLe in iteration 4.
> > > > > Reviewer pKck referred to other works in self-correction like RISE -- unfortunately, while these works have not made their code available for faithful reproduction of their method, we noted that our baselines are similar to that in RISE's paper, along with a STaR-like method. Lastly, reviewer 9mwi noted the similar objectives of our work with Inverse Constitutional AI (ICAI) in yielding constitutions from preference data; however, we noted a number of differences, including that we train models to follow said constitutions rather than to reconstruct or analyze preference annotations, our principles are on-policy to a small LM rather than generated by a frontier model, and that our method is purely automated. The reviewer's concern there is largely concerning qualitative analysis of the constitutions, their inclusion in the main text, and a comparison of the principles generated by STaPLe vs ICAI, rather than the performance (response quality) induced by said principles.
> > > > >
> > > > > ---
> > > > > We hope that this characterizes the concerns that have been raised by the other reviewers, and is helpful in guiding your confidence. Please let us know if there is anything else we can help clarify before the review period ends!

---

> > > > > > ### Comment · Reviewer_Hwww · 2025-08-07
> > > > > >
> > > > > > Thank for the authors' clarification. I think besides there's the last concern that this method should be compared with inference-time principle-based optimization methods, there are no more question. While I read other reviewers' concerns, I believe they have other aspects to examine this work and I think they are also reasonable. For me, as long as the last concern can be discussed in the revision, I think it is an acceptable work. In conclusion, I will adjust my rating to 5. Hope well.

---

### Official Review · Reviewer_pKck · 2025-07-02

**Clarity:** 3
**Significance:** 3
**Originality:** 3
**Rating:** 5
**Confidence:** 3

**Summary:**

These principles are treated as latent variables in an Expectation-Maximization-like process. At each iteration, the model generates better answers by conditioning on the gold answer (hinting), extracts candidate principles, filters them using similarity metrics (like Rouge-L or LLM judges), clusters them into a distilled constitution, and fine-tunes on the best ones. Over four iterations, the method shows consistent performance improvements on multiple benchmarks (MT-Bench, AlpacaEval, IFEval) for several open-source 7–8B models, outperforming simple self-refinement and STaR-style baselines.

**Questions:**

Please refer to the weakness for questions

**Ethical Concerns:**

["NO or VERY MINOR ethics concerns only"]

**Final Justification:**

I went through the author's response, and I am satisfied with their response to my questions.  I strongly advise the authors to include them in their updated paper. I maintain my score of "accept" for this submission.

**Limitations:**

Please address the questions in the weakness section

**Quality:**

3

**Strengths And Weaknesses:**

Strengths:
The work shows that small open models can bootstrap themselves without external teachers. Modelling principles as latent variables distinguishes the method from prior self-refinement approaches, and the principled EM derivation plus a clustering-based regulariser add technical depth . The experimental section is relatively thorough with ablations on clustering and scoring functions.

Weakness:
1.	The model gets a hint by seeing the correct answer while generating principles in the propised method.  What if this risks making the model dependent on that answer that it might just learn to copy parts of it rather than actually reason towards it.  Also, the authors say that they filter out poor candidates, but they don’t hsow how often the golden answer leaks into the principles of how much this affect the performance.
2.	The paper frames the method as EM-learning but the scoring presented in the paper doesn’t seem trainable.  Can you clarify on this?
3. the clustering step is happening after principles are generated and filtered which means, its not part of the training process if I understand it correct.  Also the clustering threshold is tuned using another optimizer?  Can you explain?
4.	The paper compares against basic self-refinement and instruction tuned models, but no results on newer methods like RISE.
5.	Do you have experiments on how the model generated principles could introduce biases and errors?

---

> ### Author Rebuttal · Authors · 2025-07-30
>
> We would like to thank Reviewer pKck for taking the time to review our work and for their valuable feedback. We appreciate that the reviewer finds the mathematical framework underlying our approach to “add technical depth”, and that the experimental section is “relatively thorough”. We address the concerns stated in the review with our responses below:
>
> \
> **1. Copying in Self-Refinement.**
> >  The model gets a hint by seeing the correct answer while generating principles in the propised method. What if this risks making the model dependent on that answer that it might just learn to copy parts of it rather than actually reason towards it. Also, the authors say that they filter out poor candidates, but they don’t hsow how often the golden answer leaks into the principles of how much this affect the performance.
>
> We would like to clarify that the model only sees the gold response when proposing a principle, not during self-refinement. As noted in the footnote on page 4, we find that the principle generated never looks like the gold response itself, in part because the prompt to propose a principle asks the model to suggest an attribute of the gold response that would make the initial response more similar to this gold target.
>
> In practice, smaller / weaker language models are not very successful at self-refinement without targeted dimensions of refinement [1,2], necessitating our filtering of all refinements that did not improve over the initial response. Note that filtering is performed on the refined responses induced by the principle and its critique, not directly on the principles themselves.
> Below, to confirm that $y^2$ does not copy $y^G$, we compute the Rouge-L precision, and report the number of samples that have a score above 0.9 — we find this number to be very small (less than 1.5% of the data), which we find to be negligible. We would like to point out that high overlap with the gold data is not an undesirable property -- after all, the gold responses are from license-friendly train sets that one would ordinarily perform SFT with -- however, we find on-policy generations to be effective.
>
> | Model  | Total Samples | \# Samples with Rouge-L Prec > 0.9 | Fraction |
> | :--------------------------: | :-------------: | :----------------------------------: | :--------: |
> | Llama-3.1 8B STaPLe Iter 1 | 28.2k         | 91  | 0.32%    |
> | Llama-3.1 8B STaPLe Iter 2 | 6.1k          | 35  | 0.57%    |
> | Llama-3.1 8B STaPLe Iter 3 | 6.3k          | 31   |  0.49%    |
> | Llama-3.1 8B StaPLe Iter 4 | 6.6k          | 41  | 0.62%    |
>
> | Model  | Total Samples | \# Samples with Rouge-L Prec > 0.9 | Fraction |
> | :--------------: | :-------------: | :-----------: | :--------: |
> | Granite-3.1 8B STaPLe Iter 1 | 24.1k         | 157  | 0.65%    |
> | Granite-3.1 8B STaPLe Iter 2 | 5.2k          | 32    | 0.61%    |
> | Granite-3.1 8B STaPLe Iter 3 | 5.9k          | 56  | 0.94%    |
> | Granite-3.1 8B StaPLe Iter 4 | 6.3k          | 87 | 1.38%    |
>
> | Model  | Total Samples | \# Samples with Rouge-L Prec > 0.9 | Fraction |
> | :---------------- | :-------------: | :------: | :--------: |
> | Qwen2.5 7B STaPLe Iter 1 | 30.9k  | 176  | 0.57%    |
> | Qwen2.5 7B STaPLe Iter 2 | 6.5k   | 53    | 0.82%    |
> | Qwen2.5 7B STaPLe Iter 3 | 7.0k | 88   | 1.25%    |
> | Qwen2.5 7B StaPLe Iter 4 | 7.1k  | 82   | 1.16%    |
>
> Since all samples in “total samples” are those where the refined generation improved over the initial response in Rouge-L F1, and the fraction of samples where the precision increased to above 0.9 did not increase substantially, it must be that the recall improved. That is, while the rate of copying does not grow much, the generated responses better cover the gold -- this is necessary for better reflecting the core elements of the gold response for the model to imitate with its on-policy generations. In works such as STaR [3], a chain-of-thought is induced that reconstructs the original gold answer — our work achieves a similar purpose in learning a distribution of on-policy generations similar to the gold, rather than overfitting to the gold.
>
> [1] Madaan et al., Self-Refine: Iterative Refinement with Self-Feedback, 2023, Thirty-seventh Conference on Neural Information Processing Systems
>
> [2] Ramji et al., Self-Refinement of Language Models from External Proxy Metrics Feedback, 2024, https://arxiv.org/abs/2403.00827
>
> [3] Zelikman et al., STaR: Bootstrapping Reasoning with Reasoning, 2022, Advances in Neural Information Processing Systems
>
> \
> **2. EM Scoring Clarification.**
> > The paper frames the method as EM-learning but the scoring presented in the paper doesn’t seem trainable. Can you clarify on this?
>
> That is partially true – the distribution over latent variables (the principle and the model response given the principle) is trainable (and we train over this), but the distribution over observed gold given the generated response is fixed and is defined via a similarity scoring function. However, this does not interfere with the EM framing of the method. The E-Step uses both distributions to generate samples from the posterior via rejection sampling, thus giving an approximate expectation over latent trajectories, while the M-Step updates just the trainable distribution to maximize the likelihood of these trajectories.
>
>
> \
> **3. Clustering in the EM Framework.**
> > the clustering step is happening after principles are generated and filtered which means, its not part of the training process if I understand it correct. Also the clustering threshold is tuned using another optimizer? Can you explain?
>
> Clustering is performed after the E-step (data generation phase) but before the M-step (model training phase), as a data augmentation procedure. A hierarchical clustering model is trained on the embeddings of the principles generated in the E-step, and the cluster representatives are chosen according to the schemes outlined in Appendix M.1 (primarily, using the cluster medoid). The self-correction trajectories generated in the E-step are augmented by replacing the principle with the cluster representative of the cluster it belongs to. The resulting trajectories (using the clustered labels) are then used for SFT training in the M-step.
>
> The results in the main paper use thresholds that we (the authors) set. However, to automate this process, we also describe and implement a Bayesian optimization procedure in Appendix M.3, which optimizes for balancing diversity across the clusters with tightness within the cluster. This method searches over threshold values, fits an Agglomerative Clustering model based on the chosen threshold for each point, and calculates this diversity-tightness objective function based on the resulting clustering, repeating iteratively by choosing next threshold value using Gaussian Process regression. In fact, we find this to perform slightly better on ablations performed on the Llama-8B and Granite-8B models.
>
>
> \
> **4. Baselines.**
> > The paper compares against basic self-refinement and instruction tuned models, but no results on newer methods like RISE.
>
> The baselines chosen are prominent methods in self-refinement and multi-iteration self-improvement, specifically those with code that is made available open-source (e.g. Self-Refine, STaR). Notably, while some recent methods like RISE and SCoRe share a broad focus on self-improvement and self-correction, neither has made their code available for reproduction, and both methods are non-trivial to implement. We consider a similar set of baselines as RISE; given our work’s premise of on-policy chain-of-thought generation given the gold response, akin to STaR, we include a STaR-like baseline.
>
>
> \
> **5. Errors and Refinement Quality.**
> > Do you have experiments on how the model generated principles could introduce biases and errors?
>
> While it was rare for intrinsic principle-based refinement to hurt response quality over the initial response (“errors”), we did find that the initial response generated by the model during self-correction became slightly worse in comparison to responses sampled from the base policy, on the basis of our Prometheus evaluation on IFEval (no degradation would be around 50%, subject to slight variance). We report this behavior in the table below:
>
> | Model  | IFEval WR (initial vs base) |
> | :---------: | :-----------: |
> | Llama-3.1 8B Constrained STaPLe Iter 1 | 44.6% |
> | Llama-3.1 8B Constrained STaPLe Iter 2 | 47.7% |
> | Llama-3.1 8B Constrained STaPLe Iter 3 | 45.3%  |
> | Llama-3.1 8B Constrained StaPLe Iter 4 | 42.6%   |
>
> | Model  | IFEval WR (initial vs base) |
> | :-------------: | :---------: |
> | Granite-3.1 8B Constrained STaPLe Iter 1 | 48.2%    |
> | Granite-3.1 8B Constrained STaPLe Iter 2 | 45.9%   |
> | Granite-3.1 8B Constrained STaPLe Iter 3 | 46.8%    |  |  |
> | Granite-3.1 8B Constrained StaPLe Iter 4 | 45.5%     |  |  |
>
> As such, comparing the refined response against generations from the base policy — rather than the initial response generated from the model during inference — was important to confirm that the quality of the refined responses generated over the iterations does indeed improve, as reflected in Tables 2 and 3. Given the refined response is what a user would follow and trust (as in recent reasoning works following a token like “Wait” [4], where our principles serve as a similar delimiter), we focus on evaluating the refinement in this work. We also compare the refined responses across iterations (“stepwise”) in Table 4, which further confirms that the refined responses do become better, before saturating, as evidenced in Figure 1 and Table 1.
>
> [4] Muennighoff et al., s1: Simple test-time scaling, 2025, https://arxiv.org/abs/2501.19393
>
> -----
>
> We hope these responses have addressed your concerns — we would be happy to address any further concerns you may have. Thank you again for your time!

---

### Official Review · Reviewer_u4GT · 2025-07-06

**Clarity:** 3
**Significance:** 1
**Originality:** 2
**Rating:** 2
**Confidence:** 4

**Summary:**

The paper proposes a framework for self-refinement of large language models.

They use the model itself prompted with a gold response to generate principles (in natural language) and critique its own initial response to a input against that generated principle, and then generate a refined response based on the critique.

These refined responses are used with supervised fine-tuning to self-improve the model to align to the self-generated principles of a good response derived from the model by looking at the gold response.

The paper poses this solution as an EM setup given the iterative self-improvement where the prompting approximates the estimation.

They also empirically arrive at clustering the generated principles (natural language) embeddings to get a reduced set for interpretability.

**Questions:**

None

**Ethical Concerns:**

["NO or VERY MINOR ethics concerns only"]

**Final Justification:**

The rebuttal fails to address flaws in Setup and Evaluation:


1. "which relies on learning explicit verbalization of principles and refinement, greatly improves" - There is no clear justification with technical analysis to support the results.

``` The results of SFT training on gold-only 50k samples reported in Table 1 in the paper, and on 90k samples in the table above, are clear indicators that our STaPLe method, which relies on learning explicit verbalization of principles and refinement, greatly improves performance over simple SFT on gold responses.```

2. The setup is mentions that it can be done with RL. The paper is purely hypothetical, and does not compare against RLHF, and the setup of regular RLHF is more cost-effective - and this work offers poorly validated.

```STaPLe is equivalent to the REINFORCE gradient for self-play under this definition. However, we view our work as a cold-start phase for such a training and we leave purely online RL training as future work. We also hypothesize that without a cold-start phase, where principles are elicited using hints and are then inserted into an un-hinted trajectory for training, the model would have taken too long (or forever) to arrive at the point of generating good/relevant principles.```

```We do not compare against existing RLHF works on constitution learning due to clear differences in setup and goals. A key difference is that we do not prescribe any principles a priori. The primary efficiency from a labor standpoint is not just in annotation, but also in curating a constitution```

I strongly do not support its acceptance, and maintain my score and initial arguments.

**Limitations:**

yes

**Quality:**

2

**Strengths And Weaknesses:**

Weakness
1. The paper is posing things as EM/posterior estimation but approximating or choosing empirical setup, the connection is not very relevant. The EM is sparsely related because its repeatedly fine-tune (maximizing step to generate critiques and responses for fine-tuning, and then the learning step, to SFT optimize with it). Figure 2.

2. Evaluation: The baseline is considered with fine-tuning on 50k gold responses but the proposed method uses the 50k with 4 additional iterations each in which 10k gold responses are used to generate critiques and responses to fine-tune with. A fair baseline would be to use all the 50k + 40k input-gold response pairs in SFT which is not considered in the paper.

3. Setup does not seem well motivated: The setup is not very useful - Why would we use Gold supervision to explicitly generate critiques vs just using SFT with them and implicitly learning it? Other than interpretability of the clustered generated set of principles, there is no reason it is not already implicit in the learning. Also, using gold annotations is more expensive - RLHF with preference annotation would be cheaper to collect. It would align the model without explicit principles. The explicit principles do not seem very useful. The paper also does not compare with RLHF.

---

> ### Author Rebuttal · Authors · 2025-07-30
>
> We would like to thank Reviewer u4GT for taking the time to review our work and for their valuable feedback. We address the concerns raised in the review through our responses below:
>
> \
> **1. EM Approach in STaPLe Algorithm.**
> > The paper is posing things as EM/posterior estimation but approximating or choosing empirical setup, the connection is not very relevant. The EM is sparsely related because its repeatedly fine-tune (maximizing step to generate critiques and responses for fine-tuning, and then the learning step, to SFT optimize with it). Figure 2.
>
> We acknowledge that our implementation approximates the EM algorithm, since exact computation of posterior marginals over latent variables is intractable in our case; However, our setup is close to what is referred to as Monte Carlo EM [1][2] – where these marginals are replaced with sample based expectations i.e. a set of latent variables, sampled from posterior, is used to optimize the gradients in the M-Step; In particular, we sample 16 latent [principles, response] pairs via rejection sampling, given current model parameters and gold responses and then maximize the likelihood over these samples.
>
> [1] William Ruth, A review of Monte Carlo-based versions of the EM algorithm, 2024 https://arxiv.org/abs/2401.00945
>
> [2] Greg C. G. Wei and Martin A. Tanner. A Monte Carlo implementation of the EM algorithm and the poor man’s data augmentation algorithms. Journal of the American Statistical Association, 85(411), 1990.
>
> \
> **2, SFT Baseline.**
> > The baseline is considered with fine-tuning on 50k gold responses but the proposed method uses the 50k with 4 additional iterations each in which 10k gold responses are used to generate critiques and responses to fine-tune with. A fair baseline would be to use all the 50k + 40k input-gold response pairs in SFT which is not considered in the paper.
>
> Thank you for this suggestion! In addition to the baseline on 50k samples we report in Table 1 (“Gold-only SFT”), here we train all 3 models on 90k samples as proposed, and compare the result against the Iteration 4 scores for STaR and STaPLe {unconstrained, constrained} in the table below:
>
> | Model                              | MT-Bench (Avg.) | AlpacaEval |
> | ---------------------------------- | --------------- | ---------- |
> | Llama 8B SFT (90k)                 | 7.50            | 28.2       |
> | Llama 8B STaR Iter 4               | 7.56            | 31.8       |
> | Llama 8B STaPLe Iter 4             | 7.71            | 33.4       |
> | Llama 8B Constrained StaPLe Iter 4 | 7.70            | 34.9       |
>
> | Model                                | MT-Bench (Avg.) | AlpacaEval |
> | ------------------------------------ | --------------- | ---------- |
> | Granite 8B SFT (90k)                 | 7.94            | 30.4       |
> | Granite 8B STaR Iter 4               | 7.96            | 35.6       |
> | Granite 8B STaPLe Iter 4             | 8.04            | 38.4       |
> | Granite 8B Constrained StaPLe Iter 4 | 8.03            | 38.8       |
>
> | Model                             | MT-Bench (Avg.) | AlpacaEval |
> | --------------------------------- | --------------- | ---------- |
> | Qwen 7B SFT (90k)                 | 7.00            | 32.3       |
> | Qwen 7B STaR Iter 4               | 7.14            | 37.8       |
> | Qwen 7B STaPLe Iter 4             | 7.24            | 40.2       |
> | Qwen 7B Constrained StaPLe Iter 4 | 7.22            | 39.9       |
>
> While SFT on 90k samples does improve slightly over the SFT on 50k samples (the results included in Table 1), it is still behind the STaR iteration 4 results, and as such, lags further behind STaPLe in both the unconstrained and constrained settings.
>
> \
> **3. Learning Principles from Human Data.**
> > Setup does not seem well motivated: The setup is not very useful - Why would we use Gold supervision to explicitly generate critiques vs just using SFT with them and implicitly learning it? Other than interpretability of the clustered generated set of principles, there is no reason it is not already implicit in the learning. Also, using gold annotations is more expensive - RLHF with preference annotation would be cheaper to collect. It would align the model without explicit principles. The explicit principles do not seem very useful. The paper also does not compare with RLHF.
>
> The results of SFT training on gold-only 50k samples reported in Table 1 in the paper, and on 90k samples in the table above, are clear indicators that our STaPLe method, which relies on learning explicit verbalization of principles and refinement, greatly improves performance over simple SFT on gold responses.
>
> As for the use of gold annotations, we do not need any new annotations for training, we use [query, response] pairs from existing open source datasets, of which there is an abundance; in STaPLe we provide a way to make the most out of those existing datasets (beyond simple SFT) without any additional annotations, with the added benefit of automatically curating a data-dependent constitution.
>
> While we train our model via an EM-based SFT framework, we agree that RL methods like PPO or GRPO could be used with the similarity score/judgment as a reward signal. In fact, in Appendix F, we show that our method is equivalent to self-play reinforcement learning by defining the advantage in terms of the difference in score against the gold response; the MC-EM gradient obtained by STaPLe is equivalent to the REINFORCE gradient for self-play under this definition. However, we view our work as a cold-start phase for such a training and we leave purely online RL training as future work. We also hypothesize that without a cold-start phase, where principles are elicited using hints and are then inserted into an un-hinted trajectory for training, the model would have taken too long (or forever) to arrive at the point of generating good/relevant principles.
>
> We do not compare against existing RLHF works on constitution learning due to clear differences in setup and goals. A key difference is that we do not prescribe any principles a priori. The primary efficiency from a labor standpoint is not just in annotation, but also in curating a constitution. Our method serves as a means to identify qualities a model should reflect in its generations; the constitutions we report in the appendix should be viewed as such a list, based on the domains and tasks included in the data mixture.
>
> Training models to follow known principles — desirable attributes for a model’s responses to reflect — is a core goal of alignment research [3, 4] at present. Our goal is to both discover such principles from the model, and train the model to follow them — implicitly learning them from gold annotations or preference data does not accomplish this task as we do not have annotations indicating *why* the gold/preferred response is desirable [5]. This motivates us to elicit principles from the data to enable human oversight of this alignment process.
>
> [3] Bai et al., Constitutional AI: Harmlessness from AI Feedback, 2022, https://arxiv.org/abs/2212.08073
>
> [4] Guan et al., Deliberative Alignment: Reasoning Enables Safer Language Models, 2025, https://arxiv.org/abs/2412.16339
>
> [5] Findeis et al., Inverse Constitutional AI: Compressing Preferences into Principles, 2025, The Thirteenth International Conference on Learning Representations.
>
> ---
> We hope these responses have addressed your concerns — if so, we would like to respectfully ask you to reconsider your assessment. We would also be happy to address any further concerns you may have. Thank you again for your time!

---

### Note · Authors · 2025-08-14

We would like to thank the reviewers who have replied for taking the time to engage with us and for providing meaningful suggestions to further improve our work. We had constructive, positive discussions with Reviewers Hwww, 9mwi, and pKck, whose reviews favor acceptance; they note the conceptual novelty, technical depth, the interpretability and flexibility in our method, and the consistent gains it yields. We commit to camera-ready changes requested during rebuttal, including greater discussion on the representativeness of constitutions and the qualitative analysis performed, ablations on the inference-time application of our principles and those yielded by related methods, and clarity-related edits. During the rebuttal period, we also completed experiments using the Qwen3 32B language model (per Reviewer Hwww's request for results with a larger LM), which showed that our method scales, even with reasoning models; these results will also be added in the camera-ready version.

However, we would like to point out that reviewer u4GT (the only reviewer recommending rejection) **did not reply to our rebuttal, nor have they submitted the mandatory acknowledgement**. Their concerns were regarding the connection to the EM algorithm, a stronger SFT baseline, and questioning why one would not just learn principles implicitly via SFT. We addressed each of these points, reported the desired experiments, and noted that having knowledge of the aspects LMs are aligned toward is among the core goals of alignment research.

We highlight the novelty of our work in using a small LM's own capabilities to uncover interpretable labels of desirable attributes to reflect, and learning the skills of principle invocation and self-correction. This fully automated self-improvement process is mathematically grounded, underlain by a Monte Carlo EM algorithm equivalent to self-play RL, and is empirically effective at boosting response quality in instruction-following without any external supervision in-the-loop.

---

### Decision · Program_Chairs · 2025-09-17

**Decision:**

Accept (poster)

**Comment:**

The paper studies the self-refinement of large language models by proposing a fully automated principle-driven self-improvement framework. Reviewers provided useful feedback, and the authors should compare to inference-time principle-based optimization methodology during the revision.